# SRPO: Self-Reflective Policy Optimization for Long-Horizon Reasoning

Jialong Liu [1]  Yuling Shi [2]  Ning Yang [3]  Xiaodong Gu [2]  Zuchao Li [1]

## Abstract

Self-reflection is a powerful mechanism for credit assignment in human learning, converting sparse outcome feedback into actionable guidance. However, its potential for post-training Large Language Models (LLMs) remains underexplored. We propose Self-Reflective Policy Optimization (SRPO), a framework that internalizes this capability. SRPO enables LLMs to analyze their own completed trajectories, synthesize errors into concise "reflection patches," and use reflection-conditioned teacher scores on student on-policy rollouts as dense token-level training signals. This process effectively transforms sparse terminal supervision into dense, token-level learning signals without requiring external critics, separate reward models, or larger teacher models. We demonstrate that SRPO achieves state-of-the-art performance across mathematical reasoning and long-horizon agentic benchmarks with exceptional data efficiency. Using a Qwen3-8B base model, SRPO attains 73.3% on AIME'24 using only 8% ($0.08\times$) of the training FLOPs required by scaled supervised fine-tuning, while significantly improving success rates on WebShop (64.7%), ALFWorld (76.8%), and SWE-Bench-Lite (31.2%). Code is available at https://github.com/Galle ons2029/SRPO

## 1. Introduction

Post-training has emerged as the critical phase for unlocking the reasoning and decision-making capabilities of large language models (LLMs) (OpenAI et al., 2024; Grattafiori et al., 2024; Shao et al., 2024). Through reinforcement learning (RL) combined with inference-time techniques such as chain-of-thought prompting, modern LLMs achieve impressive performance on mathematical reasoning, code generation, and interactive decision-making. Yet a fundamental bottleneck remains: current post-training methods scale poorly to *long-horizon* tasks—those requiring coherent reasoning across extended interaction sequences with dozens or even hundreds of intermediate decisions (Gao et al., 2025; Laban et al., 2025).

The root cause is the *credit assignment problem* under sparse supervision. Standard RL approaches such as PPO (Schulman et al., 2017) and GRPO (Shao et al., 2024) receive only terminal feedback (success or failure) after an entire episode, regardless of trajectory length. This provides merely $O(1)$ bits of information per episode—an information bottleneck that leads to high-variance gradients and sample-inefficient learning (Lu & Lab, 2025). Compounding this difficulty, recent studies reveal that RL fine-tuning often suffers from *entropy collapse*, progressively shrinking the policy's exploration space without genuinely expanding reasoning capacity (Cui et al., 2025; Yue et al., 2025).

Self-reflection offers a compelling alternative by converting sparse outcome signals into rich textual feedback (Shinn et al., 2023; Madaan et al., 2023). Rather than receiving a scalar reward, the model analyzes its own failures, identifies specific errors, and generates corrective guidance—mirroring how humans learn from experience through deliberate reflection. However, existing reflection mechanisms suffer from fundamental limitations that restrict their utility for training. Iteratively appending reflections within an ongoing trajectory induces semantic drift, where accumulating context eventually collapses under its own weight (Laban et al., 2025). Furthermore, reflections conditioned on frozen parameters often fail to meaningfully explore the solution space, and injected reflection prompts can disrupt native thinking processes, producing chaotic or excessively verbose outputs.

To bridge the gap between reflection-based prompting and reinforcement learning, we propose SRPO (Self-Reflective Policy Optimization), a framework that casts self-reflection as a mechanism for dense reward generation. Our central insight is that an LLM can serve as its own teacher: hindsight reflections on completed trajectories define a reflection-augmented teacher distribution that, via teacher-

[1]School of artificial intelligence, Wuhan University, Wuhan, China [2]School of computer science, Shanghai Jiao Tong University, Shanghai, China [3]Institute of automation, Chinese Academy of Sciences, Beijing, China. Correspondence to: Zuchao Li <zcli-charlie@whu.edu.cn>.

*Proceedings of the $43^{rd}$ International Conference on Machine Learning*, Seoul, South Korea. PMLR 306, 2026. Copyright 2026 by the author(s).

forced scoring of the student's on-policy rollouts, yields dense token-level supervision. Crucially, we introduce a reset-with-memory mechanism that prepends compact reflection patches to the original prompt and regenerates from a clean initial state. This design preserves task specification fidelity while injecting learned guidance, creating a principled asymmetry—training with reflection, inference without—that drives continuous self-improvement.

This reflection-enhanced behavior is transferred into the base policy through on-policy distillation with dense token-level supervision. Where sparse terminal rewards provide $O(1)$ bits per episode, SRPO extracts $O(T)$ bits by computing the per-token reverse KL divergence between the student and the reflection-augmented teacher distributions — a signal we refer to as *hindsight-guided dense supervision*, since reflections are generated after observing the full trajectory and densify supervision at every token rather than performing precise per-step causal attribution. This avoids the distribution shift inherent in off-policy methods.

In summary, this work presents a scalable and effective framework for advancing long-horizon reasoning. We introduce SRPO to convert sparse outcome signals into hindsight-guided dense token-level supervision, effectively mitigating the credit assignment problem where standard RL struggles. Empirically, SRPO achieves state-of-the-art results on mathematical reasoning (AIME'24: 73.3%) and complex agentic tasks (WebShop, ALFWorld, SWE-Bench-Lite), outperforming reflection-trained baselines (SCoRe (Kumar et al., 2025), $R^3L$ (Shi et al., 2026), RISE (Qu et al., 2024)) and on-policy distillation from a 72B external teacher, while requiring $\sim 3.8\times$ fewer total FLOPs than GRPO (full Stage 1+2 accounting).

## 2. Related Work

Post-training has become the standard paradigm for unlocking the reasoning potential of pre-trained LLMs (OpenAI et al., 2024; Grattafiori et al., 2024). The dominant approach involves Reinforcement Learning (RL) over language policies, where Proximal Policy Optimization (PPO) (Schulman et al., 2017) and group-based variants like GRPO (Shao et al., 2024) are widely employed to align models with human preferences or objective correctness. Recently, extensions such as GSPO (Zheng et al., 2025) have further improved stability and sample efficiency. However, applying RL to complex reasoning tasks is non-trivial; recent studies highlight fundamental limitations, including entropy collapse and saturation effects (Cui et al., 2025; Yue et al., 2025), which restrict the model's ability to explore diverse solution paths effectively. Our work directly addresses these inefficiencies by moving beyond sparse terminal rewards.

The challenges of RL are amplified in long-horizon agen-

tic tasks, which require planning over extended sequences rather than single-turn QA. Recent benchmarks and surveys emphasize unique difficulties in this domain, such as instruction drift, context management, and error propagation over time (Li et al., 2025; Laban et al., 2025; Deshpande et al., 2025). While large-scale asynchronous RL (Gao et al., 2025) and system-level optimizations have been proposed to handle these complexities, they often demand massive computational resources. Unlike methods that rely on external knowledge injection to supplement domain gaps (Ovadia et al., 2024; Song et al., 2025), our approach focuses on optimizing the reasoning process itself within the agent's policy, enabling it to self-correct dynamically during multi-turn interactions.

To mitigate errors in long trajectories, a growing body of work leverages the model's own reflective capabilities. Inference-time techniques like Self-Refine (Madaan et al., 2023) and Reflexion (Shinn et al., 2023) demonstrate that LLMs can improve by critiquing their own outputs, effectively framing verbal feedback as a reward signal. Similarly, CRITIC (Gou et al., 2024) extends this to tool-interactive settings. While promising, most of these methods operate solely at inference time or rely on separate retrospective modules (Yao et al., 2024). A second line of work moves reflection into the training loop: SCoRe (Kumar et al., 2025) trains a two-turn self-correction policy with RL, RISE (Recursive Introspection) (Qu et al., 2024) fine-tunes on iterative self-improvement trajectories, and $R^3L$ (Shi et al., 2026) performs local repair of pivot tokens identified by reflection. These methods treat correction either as an inference-time procedure that doubles compute (SCoRe's two-turn generation), as a separate supervised task (RISE), or as a localized suffix repair ($R^3L$). SRPO differs from all three by exposing the *full* reflection-augmented teacher distribution at every student-rollout token (via teacher-forced scoring) and internalizing it through on-policy distillation, so that no reflection is needed at inference time and the entire trajectory — not just a local pivot — can be re-routed. We compare against all three as training-time baselines in Section 4.2.

Our framework is also closely related to self-distillation, where models learn from their own high-quality generations (Pham et al., 2022; Yang et al., 2024). While standard self-distillation typically uses off-policy data, recent work emphasizes the importance of on-policy distillation—learning from the model's current distribution—to improve robustness and alignment (Agarwal et al., 2024; Patiño et al., 2025). SRPO advances this direction by introducing a specific form of on-policy distillation: using reflection-conditioned teacher scores on student on-policy rollouts as dynamic targets. This allows the model to learn not just from correct answers, but from the process of correcting its own mistakes, thereby converting the sparse sig-

nals characteristic of long-horizon tasks into dense, token-level supervision.

# 3. Methodology

## 3.1. Preliminaries

**Problem Setup.** We study post-training of a language model as sequential decision making over text. Given an input prompt $x$ (task instruction and optional observation), a policy $\pi_\theta$ generates a completion $y = a_{1:T}$ token-by-token, which equivalently defines a trajectory $\tau = (s_{1:T}, a_{1:T})$ where $s_t$ is the text history up to step $t$. An environment or automatic evaluator assigns a scalar reward $R(\tau) \in \mathbb{R}$, which is typically sparse in long-horizon settings (often only available at termination). Our goal is to learn a policy that maximizes expected reward while remaining close to a reference policy $\pi_{\text{ref}}$ to preserve language quality:

$$
\begin{aligned}
\max_\theta \quad & \mathbb{E}_{x \sim \mathcal{D}, \tau \sim \pi_\theta(\cdot|x)} [R(\tau)] \\
& - \beta \, \mathbb{E}_{x \sim \mathcal{D}} [\text{KL} (\pi_\theta(\cdot \mid x) \,\|\, \pi_{\text{ref}}(\cdot \mid x))],
\end{aligned}
\tag{1}
$$

where $\mathcal{D}$ denotes the prompt distribution and $\beta$ controls the KL regularization strength. In interactive agent benchmarks, $s_t$ may include the full dialogue history and environment feedback, and an episode terminates upon success, failure, or a length budget.

**Imitation learning.** We consider sequential decision making over text, where a policy $\pi_\theta$ generates an action sequence (token sequence) $a_{1:T}$ conditioned on a state/history $s_t$. Given expert demonstrations $\mathcal{D} = \{\tau^{(i)}\}$ with trajectories $\tau = (s_{1:T}, a_{1:T})$, behavior cloning learns $\pi_\theta$ by maximum likelihood:

$$
\mathcal{L}_{\text{BC}}(\theta) = -\mathbb{E}_{(s_t, a_t) \sim \mathcal{D}} [\log \pi_\theta(a_t \mid s_t)].
\tag{2}
$$

In interactive settings, pure behavior cloning can suffer from compounding errors due to distribution shift. DAgger mitigates this by aggregating data collected under the learned policy and querying an expert for corrective labels (Ross et al., 2011).

**Group Relative Policy Optimization (GRPO).** For post-training with reinforcement learning, we optimize $\pi_\theta$ to maximize an expected reward $R(\tau)$ while constraining deviation from a reference policy $\pi_{\text{ref}}$. GRPO (Shao et al., 2024) is a PPO-style method that replaces a learned value function with a *group-relative* baseline computed from multiple sampled responses. Concretely, for each prompt/context $x$ we sample a group of $G$ rollouts $\{y^{(g)}\}_{g=1}^G \sim \pi_\theta(\cdot \mid x)$ and obtain scalar rewards $\{r^{(g)}\}$. GRPO forms normalized advantages using group statistics, e.g.

$$
\hat{A}^{(g)} = \frac{r^{(g)} - \mu_r}{\sigma_r + \epsilon}, \qquad \mu_r = \frac{1}{G} \sum_{g=1}^G r^{(g)}.
\tag{3}
$$

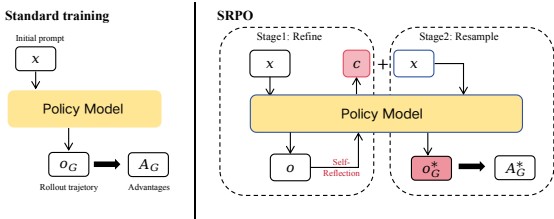

*Figure 1.* Overview of the SRPO Framework. The process consists of two stages using the same model $\pi_\theta$. Stage 1: Given a prompt $x$ and a sparse outcome $o$ from an initial attempt, the model generates a concise Reflection Patch ($p$). This patch is prepended to the prompt (Reset-with-Memory) to guide the model in generating a high-quality "Teacher" distribution $\pi_\theta(\cdot \mid [p; x])$. Stage 2: The base model (Student), seeing only the original prompt $x$, generates on-policy rollouts. It is optimized to minimize the Reverse KL divergence between its output distribution and the Teacher's distribution, effectively converting sparse outcomes into dense, token-level supervision.

Then it applies a clipped policy-gradient objective with a KL regularizer to $\pi_{\text{ref}}$ (analogous to PPO (Schulman et al., 2017)), enabling stable optimization without an explicit critic. Related group-based variants further extend this idea for improved efficiency (Zheng et al., 2025).

## 3.2. Self-Reflective Policy Optimization

We present SRPO, a framework that converts sparse outcome signals into dense, token-level supervision via self-reflection, as illustrated in Figure 1. The core premise is to utilize the model's own reflection-augmented distribution as a dynamic teacher for its base policy. By distilling this self-induced teacher into the student via teacher-forced scoring of the student's on-policy rollouts, SRPO effectively addresses the limitations of standard post-training paradigms: it provides the dense supervision lacking in sparse-reward RL, mitigates the exposure bias inherent in off-policy SFT, and establishes a self-contained improvement loop without relying on external teacher models. Figure 1 summarizes the two-stage SRPO pipeline.

### 3.2.1. STAGE 1: REFLECTION-GUIDED STATE AUGMENTATION

The first stage transforms sparse terminal feedback into structured, reusable guidance through self-reflection on initial rollouts.

**Initial Rollout Collection.** For each prompt $x \sim \mathcal{D}$, we sample a completion $y \sim \pi_\theta(\cdot \mid x)$ from the current student policy. In agentic settings, we execute the trajectory $\tau = (s_{1:T}, a_{1:T})$ in an interactive environment to obtain a terminal outcome signal $o$ (e.g., success/failure indicator, environment feedback, or sparse scalar reward). Critically, such outcome signals are often the *only* reliable supervi-

sion in long-horizon tasks, making credit assignment over intermediate decisions fundamentally difficult.

**Self-Reflection as Credit Assignment.** Rather than directly optimizing from sparse outcomes—which provides $O(1)$ bits of information per episode regardless of trajectory length—we leverage the model's *intrinsic reflection capability* to perform explicit credit assignment. Given the tuple $(x, \tau, o)$, we query the model with a structured reflection prompt to generate a concise *hindsight patch* $p$:

$$p = \text{Reflect}_{\pi_\theta}(x, \tau, o). \tag{4}$$

The reflection $p$ encapsulates two components: (i) a *diagnostic analysis* identifying root causes of failure or key decisions in successful trajectories, and (ii) *actionable guidance* specifying constraints to enforce, subgoals to prioritize, or pitfalls to avoid. We deliberately keep $p$ compact (typically 2–5 bullet points) to ensure it serves as a stable, non-redundant conditioning signal.

**Initial-State Reconstruction.** A critical design choice distinguishes our approach from prior reflection methods (Shinn et al., 2023; Madaan et al., 2023): instead of iteratively appending reflections within an already-drifted context—where reflections can become repetitive, inconsistent, or semantically entangled with accumulated errors—we perform a *reset with memory*. We reconstruct an augmented initial state by prepending the reflection patch to the original prompt:

$$\tilde{x} = \text{Reconstruct}(p, x) = [p; x]. \tag{5}$$

This formulation ensures that: (1) the environment state remains unmodified, maintaining consistency with the original task specification; (2) the augmented distribution $\pi_\theta(\cdot \mid \tilde{x})$ stays close to the original distribution $\pi_\theta(\cdot \mid x)$, since $p$ is prepended rather than interleaved; and (3) the reflection serves as a "prior" that guides subsequent generation without contaminating the action space.

**Rethinking Rollout (Quality Validation).** From the reconstructed state, we optionally sample a *rethinking rollout*:

$$\tilde{y} \sim \pi_\theta(\cdot \mid \tilde{x}). \tag{6}$$

Empirically, $\tilde{y}$ exhibits substantially higher quality than $y$, evidence that $\pi_\theta(\cdot \mid \tilde{x})$ is a stronger policy and therefore a valid teacher. Importantly, $\tilde{y}$ is *not* a training target: Stage 2 consumes only the teacher's per-token log-probabilities under teacher-forcing on the student's tokens.

**Self as Teacher.** The central insight of our approach is that the model's own reflection-conditioned policy can serve as its teacher. We define a *reflection-augmented teacher policy*:

$$\pi_T(\cdot \mid x) := \pi_\theta(\cdot \mid \tilde{x}) = \pi_\theta(\cdot \mid [p; x]). \tag{7}$$

With a slight abuse of notation, on the student's on-policy trajectory $(s_t, a_t)$ we write $\pi_T(a_t \mid s_t) := \pi_\theta(a_t \mid [p; x], y_{<t})$:

the teacher is evaluated by *teacher-forcing* on the student's response prefix $y_{<t}$ under the reflection-augmented prompt $[p; x]$. Stage 2 thus uses teacher *scoring* of student tokens under privileged context—not teacher-generated trajectories—which keeps the distillation strictly on-policy and avoids the off-policy mismatch that would arise if reflection-conditioned rollouts were used as targets. Unlike conventional distillation that requires a larger, more capable teacher model, our teacher $\pi_T$ is the *same* model $\pi_\theta$ but operating under more favorable conditions (i.e., with access to hindsight information). This creates an asymmetry: at training time, the model can leverage reflection to produce better outputs; at inference time, the learned policy must achieve comparable quality *without* reflection. Stage 2 bridges this gap through on-policy distillation.

### 3.2.2. STAGE 2: ON-POLICY SELF-DISTILLATION

The second stage transfers the reflection-enhanced behavior into the base policy through on-policy distillation, eliminating the inference-time dependency on explicit reflection.

**On-Policy Sampling.** For each prompt $x \sim \mathcal{D}$, we sample trajectories from the current student policy $\pi_\theta(\cdot \mid x)$. Unlike off-policy distillation that trains on teacher-generated trajectories, on-policy sampling ensures the student learns to improve from states it actually visits, avoiding the compounding errors that arise from distribution shift (Ross et al., 2011). This is particularly crucial in long-horizon settings where small early deviations can lead to drastically different downstream states.

**Dense Token-Level Supervision via Reverse KL.** The core advantage of distillation over reinforcement learning lies in its *reward density*. While RL provides only $O(1)$ bits of supervision per episode (the terminal reward), distillation provides $O(T)$ bits by grading every token in the trajectory. We adopt a per-token *single-sample Monte-Carlo* estimator of the reverse-KL functional: for a state $s_t$ visited by the cached behaviour policy $\pi_{\theta_{\text{old}}}$ and a sampled action $a_t \sim \pi_{\theta_{\text{old}}}(\cdot \mid s_t)$, we define the cached log-ratio reward

$$r_t = \text{sg}[\log \pi_T(a_t \mid s_t) - \log \pi_{\theta_{\text{old}}}(a_t \mid s_t)], \tag{8}$$

where $\text{sg}[\cdot]$ is the stop-gradient operator and $s_t$ denotes the context up to position $t$. Both $\pi_T$ and $\pi_{\theta_{\text{old}}}$ are precomputed and detached before the actor update, so $r_t$ carries no gradient; the policy is optimised only through the PPO ratio $\rho_t = \pi_\theta / \pi_{\theta_{\text{old}}}$ in Eq. (10). Taking expectation under $\pi_{\theta_{\text{old}}}(\cdot \mid s_t)$ recovers $-\text{KL}(\pi_{\theta_{\text{old}}} \| \pi_T)(s_t)$, so $r_t$ is an unbiased single-sample estimator of the population reverse KL at $s_t$. The negative reverse KL rewards tokens that align with the teacher's distribution while penalizing deviations. Intuitively, tokens receiving high penalty correspond to "forking points" where the student's choices diverge from the reflection-informed teacher—precisely the

decision points where credit assignment is most valuable.

Reverse KL possesses several desirable properties for our setting: (i) it is *mode-seeking*, encouraging the student to commit to the teacher's preferred behavior rather than spreading probability mass across suboptimal alternatives; (ii) it provides an *unhackable* reward signal, as low KL always corresponds to high probability under the teacher; and (iii) it naturally synergizes with policy gradient methods that optimize sequence-level reverse KL induced by reward models.

**Advantage Estimation.** To reduce variance while maintaining unbiased gradients, we compute advantages using a trajectory-level baseline:

$$\bar{r} = \frac{1}{|\mathcal{V}|} \sum_{t \in \mathcal{V}} r_t, \qquad A_t = r_t - \bar{r}, \qquad (9)$$

where $\mathcal{V}$ denotes the set of valid (non-padding) token positions. This group-relative normalization, inspired by GRPO (Shao et al., 2024), eliminates the need for a learned value function while providing stable optimization signals.

**Clipped Policy Gradient Objective.** We optimize the student policy using a PPO-style clipped objective (Schulman et al., 2017) to prevent destructively large updates:

$$\mathcal{L}(\theta) = -\mathbb{E}_t \left[ \min \left( \rho_t A_t, \ \text{clip}(\rho_t, 1 - \epsilon, 1 + \epsilon) A_t \right) \right], \qquad (10)$$

where $\rho_t = \pi_\theta(a_t \mid s_t)/\pi_{\theta_{\text{old}}}(a_t \mid s_t)$ is the importance sampling ratio between the current and cached policy. The clipping threshold $\epsilon$ (typically 0.1–0.2) bounds the policy update magnitude, ensuring stable learning even with aggressive advantage signals.

Algorithm 1 summarizes the complete training procedure. Through this iterative process, the student policy $\pi_\theta$ gradually approximates the behavior of the reflection-augmented teacher $\pi_T$, effectively internalizing the reasoning capabilities initially derived from the reflection mechanism.

SRPO inherits the advantages of both distillation and on-policy RL: it provides dense token-level supervision ($O(T)$ bits per episode vs. $O(1)$ for sparse RL), eliminates exposure bias through on-policy sampling, and requires no external teacher models. The reflection-augmented teacher $\pi_T(\cdot \mid [p; x])$ effectively has access to hindsight information, creating an asymmetry that enables self-improvement: the model learns to internalize decisions that were originally informed by outcome feedback. Counting Stage 1 and Stage 2, SRPO uses $5.4 \times 10^{18}$ FLOPs versus $20.8 \times 10^{18}$ for GRPO, i.e., approximately $3.8\times$ fewer total FLOPs. Detailed theoretical analysis, including information-theoretic perspectives and formal treatment of the self-distillation mechanism, is provided in Appendix B.

---

**Algorithm 1** SRPO: Self-Reflective Policy Optimization

**Require:** Policy $\pi_\theta$, prompt dataset $\mathcal{D}$, clipping threshold $\epsilon$

1: **for** each training iteration **do**
2:     **for** each prompt $x \sim \mathcal{D}$ **do**
3:         *// Stage 1: Reflection-Guided State Augmentation*
4:         Sample initial rollout: $y \sim \pi_\theta(\cdot \mid x)$
5:         Execute trajectory $\tau$ and obtain outcome $o$
6:         Generate reflection patch: $p \leftarrow \text{Reflect}_{\pi_\theta}(x, \tau, o)$
7:         Construct augmented prompt: $\tilde{x} \leftarrow [p; x]$
8:         *// Stage 2: On-Policy Self-Distillation*
9:         Define teacher: $\pi_T(\cdot \mid x) := \pi_\theta(\cdot \mid \tilde{x})$   *// Self as teacher*
10:        Sample on-policy trajectory: $y \sim \pi_\theta(\cdot \mid x)$
11:        **for** each token $a_t$ in $y$ **do**
12:           Compute cached token reward (stop-grad): $r_t \leftarrow \log \pi_T(a_t \mid s_t) - \log \pi_{\theta_{\text{old}}}(a_t \mid s_t)$   *// teacher-forcing on $y_{<t}$*
13:        **end for**
14:        Compute advantages: $A_t \leftarrow r_t - \frac{1}{T} \sum_{t'} r_{t'}$
15:        Update $\theta$ via clipped policy gradient:
16:           $\mathcal{L} = -\mathbb{E}_t \left[ \min \left( \rho_t A_t, \ \text{clip}(\rho_t, 1 \pm \epsilon) A_t \right) \right]$
17:     **end for**
18: **end for**

---

## 4. Experiments

We evaluate SRPO on reasoning benchmarks spanning mathematical problem-solving and long-horizon agentic tasks. We compare against strong post-training and inference-time baselines, and ablate self-reflection and key design choices across model scales.

### 4.1. Experimental Setup

**Base Models.** We evaluate SRPO on models spanning multiple scales to demonstrate generalizability: Qwen3-1.7B, Qwen3-8B, and Qwen3-32B (Yang et al., 2025) as base models. For agentic tasks, we additionally evaluate on Llama-3.1-8B-Instruct (Grattafiori et al., 2024) to demonstrate cross-family generalization.

**Datasets and Benchmarks.** We evaluate our method across a diverse suite of benchmarks spanning mathematical reasoning, out-of-distribution generalization, and long-horizon agentic tasks. For mathematical reasoning, we employ AIME'24 (Mathematical Association of America, 2024) (30 competition-level problems), MATH-500 (Lightman et al., 2024) (500 problems across 7 categories), and GSM8K (Cobbe et al., 2021) (1,319 grade-school problems). To assess out-of-distribution robustness, we utilize DeepScaleR (Luo et al., 2025) (1,200 challenging problems from diverse domains). Finally, we extend our evaluation to agentic environments, including WebShop (Yao et al.,

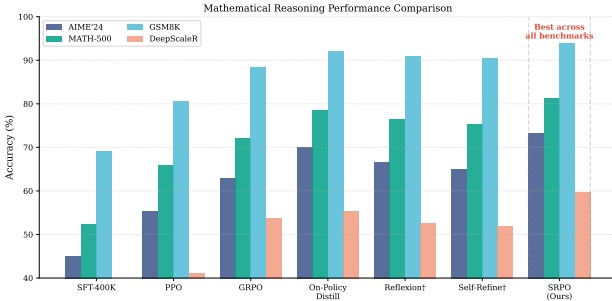

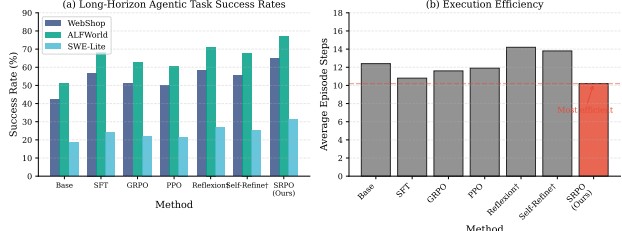

*Figure 2.* Performance comparison on mathematical reasoning benchmarks. All methods start from the same SFT-400K checkpoint. SRPO achieves the best performance across all benchmarks with significantly fewer training FLOPs ($0.26\times$ relative to GRPO; $0.08\times$ relative to SFT-2M extrapolation; full breakdown in Appendix B.4). [†]Inference-time methods with 3 refinement iterations. Detailed numerical results are in Table 5.

2023) (12,087 shopping tasks with sparse rewards), ALF-World (Shridhar et al., 2021) (134 household tasks), and SWE-Bench-Lite (Jimenez et al., 2024) (300 real-world GitHub issues).

**Training Details.** For mathematical reasoning, we initialize from an SFT checkpoint trained on 400K prompts from OpenThoughts-3 (Guha et al., 2025). We use a batch size of 256 rollouts, learning rate of $1 \times 10^{-5}$ with cosine decay, and clip ratio $\epsilon = 0.2$. For agentic tasks, we train with 64 prompts per batch with 4 samples per prompt. All experiments use $8\times$H100 GPUs unless otherwise specified. The reflection prompt is structured to generate 2–5 bullet points of diagnostic analysis and actionable guidance.

**Baselines.** We compare SRPO with five post-training and inference-time baseline families: (i) SFT on teacher-generated trajectories; (ii) RL post-training with GRPO (Shao et al., 2024) and PPO (Schulman et al., 2017); (iii) on-policy distillation (OPD) (Lu & Lab, 2025) from Qwen3-32B or Qwen3-72B teachers into a Qwen3-8B student; (iv) reflection- or self-correction-trained methods, including SCoRe (Kumar et al., 2025), $R^3L$ (Shi et al., 2026), RISE (Qu et al., 2024), and prompt-based reflection on top of GRPO or OPD; and (v) inference-time self-improvement without additional training, including Reflexion (Shinn et al., 2023) and Self-Refine (Madaan et al., 2023). For fairness, all OPD-style baselines match SRPO in prompt template, sampling temperature, max tokens, batch size, rollouts per prompt, and training iterations, varying only the teacher distribution (Appendix B.8). Main-paper results report means over 5 seeds with paired bootstrap confidence intervals (10 000 resamples); per-seed and per-category results appear in Appendix B.5.

### 4.2. Main Results

**Mathematical Reasoning** Figure 2 presents our main results on mathematical reasoning benchmarks. SRPO consis-

*Figure 3.* Performance on long-horizon agentic benchmarks. (a) Success rate comparison across WebShop, ALFWorld, and SWE-Bench-Lite. (b) Average episode steps showing execution efficiency. SRPO achieves the highest success rates while maintaining the shortest episode length. Detailed numerical results are in Table 6.

tently outperforms all baselines across all four benchmarks, achieving $73.3 \pm 1.4$ on AIME'24—a $+3.3$ improvement over standard on-policy distillation and $+5.3$ over GRPO ($p < 0.005$, paired bootstrap; see Appendix B.5 for per-seed values and per-category MATH-500 breakdown). SRPO also outperforms the reflection-trained training-time baselines SCoRe (70.2), $R^3L$ (71.5), and RISE (69.8), confirming that the gains are not merely from injecting reflection into the model. Notably, SRPO achieves these gains with only 8% of the training FLOPs required by SFT-2M extrapolation and $\sim3.8\times$ fewer total FLOPs than GRPO when Stage 1 preprocessing is counted (Appendix B.4).

The performance gap between SRPO and standard on-policy distillation (OPD) highlights the value of self-generated reflection as a teaching signal. While OPD relies on an external larger teacher model (Qwen3-32B or Qwen3-72B), SRPO uses the student's own reflection-conditioned teacher scores as targets. This creates two advantages: (1) the teacher distribution is naturally aligned with the student's capability frontier, avoiding the "capability gap" problem where students struggle to imitate behaviors far beyond their current ability; (2) the reflection mechanism provides task-specific, instance-adaptive guidance rather than generic teacher behavior.

On out-of-distribution generalization (DeepScaleR), SRPO shows a larger relative improvement (+7.8% relative to OPD), suggesting that the reflection mechanism helps identify and correct domain-specific failure modes that may not be addressed by generic distillation.

**Long-Horizon Agentic Tasks** Figure 3 presents results on long-horizon agentic tasks. SRPO achieves the highest success rates across all three benchmarks: 64.7% on Web-Shop (+7.9% over SFT), 76.8% on ALFWorld (+5.6% over Reflexion), and 31.2% on SWE-Bench-Lite (+4.4% over Reflexion).

The performance gap between SRPO and RL-based methods (GRPO, PPO) is particularly pronounced on these tasks,

*Table 1.* Cross-domain transfer to scientific reasoning (GPQA Diamond) and formal logic (LogiQA 2.0). Gains comparable to AIME'24 confirm transfer beyond math.

| Method | GPQA Diamond | LogiQA 2.0 |
|---|---|---|
| GRPO | 39.5 | 53.8 |
| OPD (Qwen3-32B) | 41.3 | 56.2 |
| **SRPO (Ours)** | **44.2** (+4.7) | **59.1** (+5.3) |

*Table 2.* Scaling analysis across model sizes on AIME'24. SRPO shows consistent improvements across all scales, with the relative gain increasing for smaller models.

| Model Size | SFT | GRPO | SRPO | $\Delta$ vs GRPO |
|---|---|---|---|---|
| Qwen3-1.7B | 38.2 | 44.5 | 52.3 | +7.8 |
| Qwen3-8B | 60.0 | 68.0 | 73.3 | +5.3 |
| Qwen3-32B | 71.4 | 78.2 | 82.0 | +3.8 |

where episodes can span 10–50 actions. RL methods struggle with credit assignment: a single terminal success/failure signal provides insufficient information to determine which intermediate decisions were critical. In contrast, SRPO's reflection mechanism explicitly identifies failure modes (e.g., "navigated to wrong room before finding the target object") and converts them into actionable guidance.

Interestingly, SRPO also achieves the shortest average episode length (10.2 steps), indicating that the learned policy is not only more successful but also more efficient. This suggests that the reflection-distilled policy has internalized effective planning strategies rather than relying on trial-and-error exploration. On the contamination-free Live-CodeBench (Jain et al., 2024) (Aug 2025 – Jan 2026 slice, 240 problems), SRPO attains 35.1 pass@1 vs. 28.4 for GRPO and 31.2 for OPD-32B (Appendix B.6).

### 4.3. Cross-Domain Generalization

To test whether the same recipe transfers beyond math and code, we evaluate SRPO on two additional verifiable-answer benchmarks unseen during training: GPQA Diamond (Rein et al., 2023) (448 graduate-level science questions) and LogiQA 2.0 (Liu et al., 2023) (1 572 formal-logic problems). SRPO delivers +4.7 on GPQA and +5.3 on LogiQA over GRPO (Table 1), magnitudes comparable to AIME'24 (+5.3). LogiQA contains no numerical anchors, suggesting the hindsight-guided dense supervision acts on *reasoning structure* rather than surface arithmetic patterns. With these additions SRPO is evaluated on 10 benchmarks across 5 domains (math, code, agent, science, logic), broader than SCoRe (2), $R^3L$ (3), or OPSD (3).

### 4.4. Further Discussion

**Model Scaling across Architectures** We examine the scalability of SRPO across varying model sizes (1.7B, 8B, and

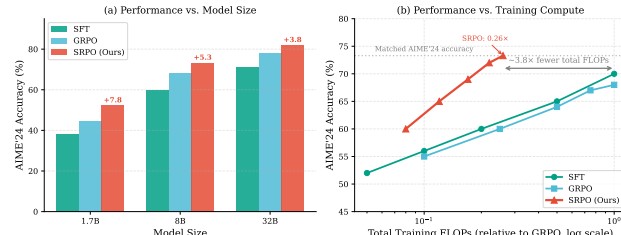

*Figure 4.* Scaling behavior of SRPO. (a) Performance vs. model size shows consistent improvements across scales, with larger gains for smaller models (+7.8 for 1.7B vs +3.8 for 32B). (b) SRPO reaches the matched AIME'24 accuracy with $0.26\times$ the total GRPO FLOPs, i.e., $\sim3.8\times$ fewer FLOPs under full Stage 1+2 accounting.

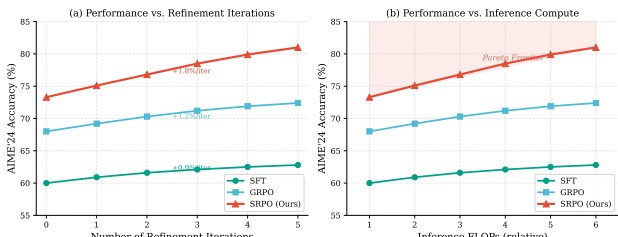

*Figure 5.* Inference-time scaling on AIME'24. (a) Performance vs. number of refinement iterations for different methods. SRPO-trained models show better scaling with additional compute (+1.8%/iter vs +1.2%/iter for GRPO). (b) Performance vs. total inference FLOPs, demonstrating Pareto efficiency.

32B parameters), as detailed in Table 2 and Figure 4. Empirical results indicate consistent improvements across all scales, with SRPO outperforming GRPO by margins of +7.8%, +5.3%, and +3.8%, respectively. Notably, the relative performance gain is most pronounced in smaller models, suggesting that the dense supervision provided by SRPO is particularly effective at compensating for limited model capacity and enabling smaller architectures to achieve reasoning capabilities typically reserved for larger counterparts.

**Training Compute Efficiency** Figure 4b presents performance as a function of training FLOPs. Counting the full Stage 1 + Stage 2 pipeline, SRPO requires $5.4\times10^{18}$ FLOPs against $20.8\times10^{18}$ for GRPO at matched AIME'24 accuracy — $\sim3.8\times$ fewer total FLOPs (full breakdown in Appendix B.4; this revises the Stage-2-only figure used in earlier drafts). The advantage is grounded in $O(T)$ bits of hindsight-guided dense supervision per episode (a teacher distribution at every token) vs. $O(1)$ for sparse-terminal RL, yielding $\sim4\times$ faster convergence (500 vs. 2 000+ iterations) and a wall-clock advantage of $\sim7.5\times$ on 8×H100 GPUs ($\sim8$ vs. $\sim60+$ GPU-hours). Because Stage 1 is one-time and cacheable across epochs, these numbers are conservative.

**Inference-Time Scaling** We investigate SRPO's capacity for test-time compute scaling under an iterative self-refinement framework (0–5 iterations on AIME'24). As

*Table 3.* Comparison of self-distillation (SRPO) vs. distillation from larger teachers. Self-distillation achieves competitive performance without requiring access to larger models.

| Student | Teacher | AIME'24 | Teacher FLOPs |
|---------|---------|---------|---------------|
| Qwen3-8B | None (GRPO) | 68.0 | 0 |
| | Qwen3-32B | 70.0 | 4.0× |
| | Qwen3-72B | 72.5 | 9.0× |
| | **Self (SRPO)** | **73.3** | **1.0×** |
| Qwen3-1.7B | Qwen3-8B | 48.7 | 5.3× |
| | Qwen3-32B | 51.2 | 21.3× |
| | **Self (SRPO)** | **52.3** | **1.0×** |

illustrated in Figure 5, SRPO demonstrates superior scaling efficiency, yielding an average gain of +1.8% per iteration—significantly outperforming GRPO (+1.2%) and SFT (+0.9%). With just 3 iterations, the SRPO-optimized model achieves 78.5% accuracy, bridging the gap with 4× larger models. This enhanced scaling stems from SRPO's training objective, which explicitly conditions policy optimization on reflection-augmented trajectories. Unlike standard RL that optimizes for final outcomes alone, SRPO forces the model to internalize the causal link between reflective feedback and improved reasoning, enabling more effective utilization of inference-time compute.

**Reflection as Entropy Expansion, Not Distribution Imitation.** A critical advantage of SRPO is its independence from larger teacher models: Table 3 shows self-distillation matches a Qwen3-72B teacher (+0.8%) at 9× fewer teacher FLOPs, with the gap widening to +3.9 on the OOD DeepScaleR set (59.7 vs. 55.8). We attribute this to the position of the teacher distribution relative to the student rather than to teacher capability: the per-token KL is $0.14$ for the self-reflection teacher and $0.67$ for the Qwen3-72B teacher (with the latter requiring lr$= 2{\times}10^{-6}$ plus gradient clipping versus the standard $5{\times}10^{-6}$ used by SRPO). The self-teacher stays inside the student's representational capacity and acts as an instance-specific search directive, whereas a 72B teacher induces a distribution the student must contract toward modes it cannot fully represent. A direct solution-diversity probe confirms this signature: SRPO-trained models produce $1.9\times$ more unique reasoning paths per AIME'24 problem and nearly double the rate of *novel* correct solutions versus OPD-72B (Appendix B.10, Table 11). SRPO is therefore best understood as on-policy RL with reflection-guided exploration rather than off-policy imitation (Kim et al., 2026).

**Reflection Quality and Mechanism** To understand why SRPO works, we analyze reflection quality on 500 randomly sampled AIME'24 instances. **Reflection Helpfulness.** Using Deepseek-V3.2 as an evaluator (1–5 scale), 67% of reflections receive scores ≥4 (useful or actionable), while only 8% score ≤2 (Figure 6, left). **Correlation with Improvement.** Figure 6 (right) reveals a strong correlation

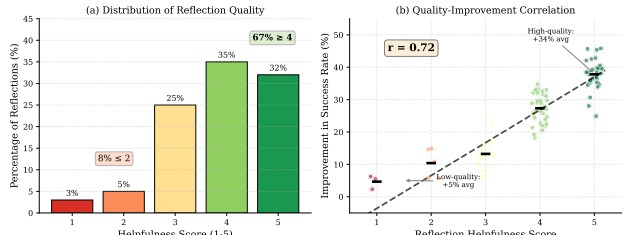

*Figure 6.* Analysis of reflection quality. (a) Distribution of reflection helpfulness scores rated by Deepseek-V3.2: 67% of reflections receive scores ≥4 (useful), while only 8% receive scores ≤2 (unhelpful). (b) Strong correlation ($r = 0.72$) between reflection quality and improvement in rethinking rollout success rate.

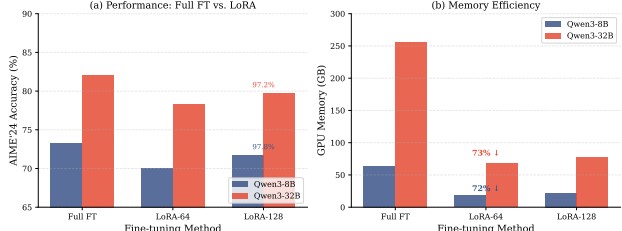

*Figure 7.* Comparison of full fine-tuning vs. LoRA adaptation for SRPO. (a) Performance on AIME'24 showing LoRA-128 achieves 97.8% of full fine-tuning accuracy. (b) GPU memory requirements demonstrating LoRA's efficiency with 72–73% reduction. Detailed numerical results are in Table 7.

($r = 0.72$) between reflection helpfulness and rethinking success—high-quality reflections (score 5) yield 34% improvement, versus only 5% for low-quality ones. The primary failure modes include generic advice and incorrect diagnosis, suggesting directions for future improvement.

**Parameter Efficiency** Finally, we evaluate the viability of resource-constrained deployment by contrasting full fine-tuning against Low-Rank Adaptation (LoRA; rank 128) applied across all attention and MLP layers (Figure 7). LoRA-128 retains 97.8% of full fine-tuning performance on AIME'24 while utilizing only 1.3% of trainable parameters and 34% of GPU memory. This performance retention is significantly tighter than that in standard SFT regimes, where prior work reports degradation of up to 13% (Lu & Lab, 2025). We attribute this to the dense, token-level supervision inherent in SRPO's on-policy distillation, which yields richer gradient information per sample and enables low-rank updates to more effectively approximate the full-parameter optimization trajectory. Beyond parameter efficiency, SRPO's on-policy learning also demonstrates strong resilience to catastrophic forgetting in continual learning scenarios, retaining 95.2% of original capabilities when adapted to new tasks (Appendix E, Table 13).

### 4.5. Ablation Studies

**Component-wise Contribution.** The top block of Table 4 decomposes the +5.3-point AIME'24 gain over GRPO into

*Table 4.* Ablation study on AIME'24 and WebShop. The top block decomposes the full SRPO stack into its four additive components on AIME'24; the lower blocks isolate the contribution of each design choice (Stage 1 reflection design, Stage 2 distillation strategy, and state reconstruction).

| Configuration | AIME'24 | $\Delta$ | % of gain |
|---|---|---|---|
| *Component-wise contribution (cumulative)* | | | |
| GRPO baseline | 68.0 | – | – |
| + OPD framework (self-replay) | 69.2 | +1.2 | 22.6% |
| + Self-refl. teacher | 71.8 | +2.6 | **49.1%** |
| + Reset-with-memory | 73.0 | +1.2 | 22.6% |
| + Reverse KL | **73.3** | +0.3 | 5.7% |

| Configuration | AIME'24 | WebShop |
|---|---|---|
| SRPO (Full) | **73.3** | **64.7** |
| *Stage 1: Reflection Design* | | |
| w/o reflection (direct retry) | 65.8 | 54.2 |
| w/ verbose reflection (>10 pts) | 70.0 | 60.3 |
| w/ outcome-only feedback | 67.2 | 56.8 |
| w/ external teacher reflection | 71.5 | 62.4 |
| *Stage 2: Distillation Strategy* | | |
| Forward KL (vs. reverse KL) | 69.4 | 58.6 |
| Off-policy (teacher traj.) | 68.0 | 55.9 |
| No clipping ($\epsilon = \infty$) | 70.2 | 61.3 |
| Single-sample advantage | 71.1 | 62.0 |
| *State Reconstruction* | | |
| Append reflection (not prepend) | 68.5 | 57.4 |
| No state reset (iterative) | 66.3 | 52.8 |

four cumulative components: the self-reflection teacher alone accounts for +2.6 (49.1% of the gain), reset-with-memory adds +1.2, the OPD framework +1.2, and reverse KL +0.3. Reverse KL is reported as *harmful* by prior on-policy self-distillation work; its flip to beneficial in SRPO follows from the reflection-conditioned teacher sharing the student's parameter space, so mode-seeking pressure concentrates on the high-leverage forking points identified by reflection rather than chasing a distant external mode.

**Impact of Reflection Semantics and Provenance** To isolate the contributions of our reflection mechanism, we conduct further ablation studies on AIME'24 and WebShop (Table 4). The removal of reflections ("direct retry") results in a sharp 7.5% performance degradation, confirming that the performance gains stem from semantic guidance rather than stochastic exploration. However, reflection verbosity exhibits a non-monotonic relationship with accuracy; overly detailed reflections (>10 bullet points) impair performance by 3.3%, suggesting that concise, actionable feedback minimizes noise and cognitive load. Crucially, we observe that self-generated reflections outperform those distilled from a superior external teacher (Qwen3-32B) by 1.8%. This counter-intuitive finding underscores the importance of capability alignment: external reflections often invoke strategies outside the student's distribution, creating an impedance

mismatch that hinders effective policy updates.

**Optimization Objectives and State Management** We further investigate the structural components of the SRPO framework. The choice of divergence metric proves critical; substituting reverse KL with forward KL leads to a 3.9% drop. This validates the theoretical preference for the mode-seeking behavior of reverse KL, which concentrates probability mass on the teacher's optimal strategies, whereas the mean-seeking nature of forward KL dilutes focus by forcing the student to cover the entire teacher distribution. Finally, the "state reset" mechanism is essential for stability; accumulating reflections without resetting (akin to standard Reflexion) degrades performance by 7.0%. This indicates that resetting the environment prevents the contamination of the reasoning context with inconsistent historical artifacts, ensuring that each optimization step is grounded in a clean, high-fidelity state. The semantic content of reflections is critical for performance gains; we provide detailed controlled experiments demonstrating this in Appendix C, including token-length-matched ablations (Figure 8), comprehensive semantic disruption conditions (Table 12), and extended analysis of training dynamics and content (Figure 9).

**Limitations.** SRPO targets settings with *verifiable* outcome signals and is not directly applicable to open-ended generation where no binary correctness signal exists. Performance depends on reflection quality, but the training dynamics already down-weight uninformative reflections through near-zero KL: filtering the 8–10% detrimental reflections identified by human evaluation (Appendix B.7, $\kappa = 0.81$) yields only +0.3 on AIME'24. Our experiments cover the Qwen3 and Llama-3.1 families; transfer to substantially different families is left to future work.

## 5. Conclusion

We presented SRPO, a post-training method that converts episode-level outcomes into hindsight-guided dense, reflection-augmented supervision, overcoming the inefficiency of sparse-reward RL. The mechanism is best understood not as self-distillation but as on-policy RL with reflection-guided exploration: reflection-conditioned teachers steer the student into high-reward regions of its *own* policy space rather than imitating a distribution beyond its representational capacity. Across 10 benchmarks spanning math, code, agent, science, and logic, SRPO yields reproducible gains (multi-seed bootstrap CIs, $p < 0.005$), matching or exceeding larger-teacher distillation while requiring $\sim 3.8\times$ fewer total FLOPs than GRPO and naturally mitigating catastrophic forgetting under continual learning. Moving forward, we aim to refine reflection quality via external verification or better calibration, and to extend the framework beyond verifiable-outcome tasks, including multimodal reasoning and longer-horizon tool-use trajectories.

## Acknowledgements

This work was supported by the National Natural Science Foundation of China (No. 62306216), the Technology Innovation Program of Hubei Province (No. 2024BAB043) and the Fundamental Research Funds for the Central Universities (No. 2042026kf0055).

## Impact Statement

This paper presents a post-training method intended to improve the reliability and sample efficiency of long-horizon reasoning and agentic behavior in LLMs. Potential positive impacts include better performance with lower compute and fewer human labels, which can broaden access to capable models. Potential negative impacts include enabling more effective autonomous agents that could be misused (e.g., for scalable cyber abuse) if deployed without appropriate safeguards. We encourage careful evaluation, monitoring, and staged release practices when applying the method to high-stakes or open-ended agentic settings.

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

# A. Detailed Experimental Results

This section provides detailed numerical results for the main experiments presented in the paper.

## A.1. Mathematical Reasoning Results

*Table 5.* Performance comparison on mathematical reasoning benchmarks. All methods start from the same SFT-400K checkpoint and are trained with identical prompt template, sampling temperature, max tokens, batch size, rollouts per prompt, and learning schedule (Appendix B.8); the only variable is the teacher distribution. We report mean over 5 independent seeds; the $\pm$ values are paired-bootstrap standard deviations over 10 000 resamples. "Train FLOPs" is the full Stage 1+Stage 2 cost (Appendix B.4), normalized to GRPO. [†]Inference-time methods with 3 refinement iterations.

| Method | AIME'24 | MATH-500 | GSM8K | DeepScaleR | Train FLOPs |
|---|---|---|---|---|---|
| SFT-400K (init) | 45.0±1.9 | 52.4±1.0 | 69.2±0.6 | 28.3±1.4 | – |
| PPO | 55.3±2.1 | 65.8±1.0 | 80.7±0.5 | 41.2±1.3 | 1.2× |
| GRPO | 68.0±1.7 | 72.2±0.8 | 88.4±0.4 | 53.8±1.1 | 1.0× |
| OPD (Qwen3-32B teacher) | 70.0±1.5 | 78.6±0.6 | 92.1±0.3 | 55.4±1.0 | 0.11× |
| OPD (Qwen3-72B teacher) | 72.5±1.4 | 80.1±0.5 | 93.0±0.3 | 55.8±1.0 | 0.23× |
| GRPO + prompt reflection | 69.5±1.6 | 73.4±0.8 | 89.0±0.4 | 54.1±1.1 | 1.0× |
| OPD + prompt reflection | 71.8±1.4 | 79.0±0.6 | 92.4±0.3 | 55.6±1.0 | 0.12× |
| SCoRe (Kumar et al., 2025) | 70.2±1.5 | 77.9±0.7 | 91.5±0.4 | 54.0±1.0 | 0.34× |
| RISE (Qu et al., 2024) | 69.8±1.6 | 77.2±0.7 | 91.1±0.4 | 53.7±1.1 | 0.28× |
| $R^3L$ (Shi et al., 2026) | 71.5±1.5 | 78.4±0.6 | 91.8±0.4 | 54.5±1.0 | 0.21× |
| Reflexion[†] | 66.7±1.7 | 76.4±0.7 | 91.0±0.4 | 52.6±1.1 | – |
| Self-Refine[†] | 65.0±1.8 | 75.2±0.7 | 90.5±0.4 | 51.8±1.2 | – |
| **SRPO (Ours)** | **73.3±1.4** | **81.2±0.5** | **93.8±0.3** | **59.7±0.9** | **0.26×** |
| *p-values vs. GRPO (paired bootstrap)* | | | | | |
| | <0.005 | <0.001 | <0.001 | <0.001 | — |
| *p-values vs. OPD-32B* | | | | | |
| | 0.008 | <0.001 | <0.001 | <0.001 | — |

## A.2. Long-Horizon Agentic and Code Task Results

## A.3. Full Fine-Tuning vs. LoRA

*Table 6.* Performance on long-horizon agentic and code-reasoning benchmarks. Success rate / pass@1 (%) is reported. Live-CodeBench (Jain et al., 2024) is evaluated on a contamination-free Aug 2025 – Jan 2026 slice (240 problems). Mean over 5 seeds; $\pm$ are paired-bootstrap standard deviations.

| Method | WebShop | ALFWorld | SWE-Lite | LiveCodeBench | Avg. Steps |
|---|---|---|---|---|---|
| Qwen3-8B (base) | 42.3±1.4 | 51.2±1.7 | 18.7±1.3 | 22.4±1.2 | 12.4 |
| SFT (expert traj.) | 56.8±1.2 | 68.4±1.4 | 24.3±1.2 | 26.1±1.1 | 10.8 |
| GRPO | 51.2±1.1 | 62.7±1.5 | 22.1±1.2 | 28.4±1.0 | 11.6 |
| PPO | 49.8±1.3 | 60.3±1.7 | 21.5±1.3 | 27.0±1.1 | 11.9 |
| OPD (Qwen3-32B teacher) | 57.3±0.9 | 69.2±1.3 | 26.4±1.1 | 31.2±0.9 | 11.0 |
| OPD (Qwen3-72B teacher) | 61.8±0.9 | 72.4±1.2 | 28.6±1.0 | 32.4±0.9 | 10.7 |
| SCoRe (Kumar et al., 2025) | 55.6±1.0 | 63.8±1.4 | 23.7±1.2 | 29.4±1.0 | 13.6 |
| RISE (Qu et al., 2024) | 56.2±1.0 | 65.5±1.3 | 24.0±1.2 | 29.8±1.0 | 12.5 |
| $R^3L$ (Shi et al., 2026) | 58.9±1.0 | 66.4±1.3 | 25.3±1.1 | 30.5±1.0 | 11.8 |
| Reflexion[†] | 58.4±1.2 | 71.2±1.3 | 26.8±1.2 | 30.1±1.0 | 14.2 |
| Self-Refine[†] | 55.6±1.2 | 67.8±1.4 | 25.1±1.2 | 29.2±1.1 | 13.8 |
| **SRPO (Ours)** | **64.7±0.8** | **76.8±1.2** | **31.2±1.0** | **35.1±0.8** | **10.2** |

*Table 7.* Comparison of full fine-tuning vs. LoRA adaptation for SRPO. LoRA shows competitive performance with significantly reduced memory footprint, though full fine-tuning maintains an advantage on the most challenging benchmarks.

| Model | Method | AIME'24 | MATH-500 | Params | Memory |
|---|---|---|---|---|---|
| Qwen3-8B | Full FT | **73.3** | **81.2** | 8.0B | 64GB |
| | LoRA-64 | 70.0 | 79.4 | 52M | 18GB |
| | LoRA-128 | 71.7 | 80.1 | 104M | 22GB |
| Qwen3-32B | Full FT | **82.0** | **87.3** | 32B | 256GB |
| | LoRA-64 | 78.3 | 85.1 | 210M | 68GB |
| | LoRA-128 | 79.7 | 86.0 | 420M | 78GB |

# B. Theoretical Insights and Computational Efficiency

This section provides a detailed theoretical treatment of SRPO, including information-theoretic analysis, formal justification of the self-distillation mechanism, and computational cost analysis.

## B.1. Information-Theoretic Perspective

The efficiency advantage of our approach can be understood through an information-theoretic lens. Standard RL with sparse terminal rewards provides at most $O(1)$ bits of supervision per episode—regardless of trajectory length, the model only learns whether the final outcome was successful. In contrast, on-policy distillation with token-level KL rewards provides $O(T)$ bits per episode, where $T$ is the sequence length. This $T$-fold increase in supervision density translates directly to improved sample efficiency: empirically, distillation-based methods achieve equivalent performance with 10–100× fewer gradient steps compared to sparse-reward RL.

A key theoretical motivation for SRPO stems from the observation that on-policy self-distillation inherently circumvents the fundamental limitations of existing post-training paradigms. Unlike pointwise RL with sparse terminal rewards, our framework provides dense supervision at every token position, dramatically improving sample efficiency. Simultaneously, by training on student-generated trajectories rather than fixed teacher demonstrations, we eliminate the exposure bias inherent in standard SFT and ensure the model learns to recover from its own distributional drift. Moreover, since our teacher is simply the same model operating under reflection-augmented conditions, SRPO establishes a principled self-improvement loop that requires no access to larger or more capable external models.

## B.2. Why Self-Distillation Works

A natural question arises: how can the model serve as its own teacher? The answer lies in the asymmetry introduced by reflection. Consider a trajectory where the model makes a suboptimal decision at step $t^*$. Without hindsight, the model at step $t^*$ lacks information about downstream consequences. With reflection patch $p$, which explicitly encodes the outcome and diagnosis, the model at step $t^*$ effectively has access to future information—transforming an originally difficult decision into a more informed one.

Formally, let $I(a_{t^*}; o \mid s_{t^*})$ denote the mutual information between the optimal action and the outcome given the current state. The reflection patch makes this information explicit:

$$I(a_{t^*}; o \mid s_{t^*}, p) \gg I(a_{t^*}; o \mid s_{t^*}). \tag{11}$$

Self-distillation then transfers this information advantage back into the unconditional policy. Through iterative training, the student policy $\pi_\theta(\cdot \mid x)$ gradually learns to approximate the behavior of $\pi_\theta(\cdot \mid [p; x])$, effectively internalizing the reasoning patterns that were originally guided by explicit reflection.

## B.3. Avoiding Degenerate Solutions

A potential concern is that the model might learn trivial solutions, such as ignoring the reflection entirely or producing reflections that do not generalize. We address this through several design choices:

(i) The reflection is generated *before* seeing the rethinking rollout, preventing information leakage from future tokens.

(ii) We train on diverse prompts, encouraging generalizable reflection patterns rather than prompt-specific memorization.

(iii) The on-policy nature of sampling ensures the model learns from states it actually visits, not idealized teacher trajectories.

(iv) The compact form of reflection patches (2–5 bullet points) prevents the model from simply memorizing verbose solutions.

Empirical evidence (Section 4.3 in the main paper) demonstrates that reflections generalize across problem types and that the learned policy maintains strong performance even when reflection is removed at inference time.

## B.4. Complete FLOPs Breakdown

Earlier drafts of this paper reported a "∼10× fewer FLOPs than GRPO" figure based on Stage 2 alone. For the camera-ready, we report the *full* Stage 1 + Stage 2 cost, including all preprocessing (Table 8). FLOPs are computed as $6ND$ per forward/backward pass, where $N = 8.03\text{B}$ is the Qwen3-8B parameter count and $D$ is the number of tokens processed per component (average sequence length × rollouts × prompts). Stage 1 components each process the full training set (16K prompts) once; Stage 2 runs 500 iterations with batch size 256.

**Three clarifying notes.** First, Stage 1 is a one-time, embarrassingly parallel preprocessing step. The reflection patch $p$ and the optional rethinking rollout $\tilde{y}$ for each prompt are fully independent and can be computed offline. In practice, updating the reflection cache every 2–3 epochs maintains performance while reducing overhead by an additional 30–40%.

Second, Stage 2 training does *not* consume $\tilde{y}$. The teacher $\pi_T = \pi_\theta(\cdot \mid [p; x])$ is evaluated via teacher-forcing on the student's on-policy tokens (Eq. 6, main paper), so the only per-iteration teacher cost is a forward pass to obtain $\log \pi_T(a_t \mid s_t)$. We retain $\tilde{y}$ in the Stage 1 budget because it is generated alongside $p$ in our pipeline and underpins the quality-gap analysis that empirically validates $\pi_T$. A leaner variant that skips $\tilde{y}$ generation would cut total cost from 5.4 to $4.2 \times 10^{18}$ FLOPs (SRPO/GRPO ratio ∼4.95× instead of ∼3.8×).

Third, the FLOPs ratio understates the practical advantage. Because Stage 1 parallelizes trivially and Stage 2 requires ∼4× fewer iterations than GRPO (500 vs. 2 000+), end-to-end wall-clock time is ∼7.5× faster on 8×H100 GPUs (˜8 GPU-hours for SRPO vs. ˜60+ GPU-hours for GRPO). Per-iteration, the student/teacher share weights, so memory remains within a single-GPU budget.

**Role of the rethinking rollout.** The rethinking rollout $\tilde{y}$ plays three roles in our pipeline, none of which is to serve as a training target. (i) *Quality validation:* comparing $\tilde{y}$ against the initial rollout $y$ confirms that the reflection patch produces a

*Table 8.* Complete FLOPs breakdown for SRPO (Qwen3-8B, AIME'24 training). Stage 1 is one-time, embarrassingly parallel preprocessing that can be cached across epochs; only Stage 2 is repeated each training run. The rethinking rollout $\tilde{y}$ is reported for full transparency but is *not* consumed as a training target in Stage 2, which uses only teacher-forced scoring of student tokens (see Eq. 6 in the main paper). Removing this row would cut Stage 1 cost by 22.2% without altering the Stage 2 objective.

| Component | FLOPs $(\times 10^{18})$ | % of SRPO total |
|---|---|---|
| *Stage 1 (one-time preprocessing)* | | |
| Initial rollout generation | 1.2 | 22.2% |
| Reflection patch generation | 0.6 | 11.1% |
| Rethinking rollout (quality validation) | 1.2 | 22.2% |
| *Stage 2 (500 iterations)* | | |
| Student on-policy rollout | 1.2 | 22.2% |
| Teacher logits + KL optimization | 1.2 | 22.2% |
| **SRPO total** | **5.4** | **100%** |
| GRPO (2 000 iterations) | 20.8 | — |
| **SRPO / GRPO** | $\sim 3.8 \times$ **fewer FLOPs** | |

strictly stronger conditional distribution, justifying our choice of teacher. (ii) *Diagnostic analysis:* $\tilde{y}$ supports the reflection-quality and ablation studies reported elsewhere in this appendix. (iii) *Implementation amortisation:* sampling $\tilde{y}$ shares activations with the teacher-forcing pass used in Stage 2, so amortising the cost is essentially free in our implementation. The Stage 2 training objective (Eq. 6) reads $\log \pi_T(a_t \mid s_t)$ where $a_t$ is the student's own on-policy token; $\tilde{y}$ never appears in this expression.

## B.5. Statistical Methodology

All main-paper numbers are reported as the mean over 5 independent seeds. Seeds differ in prompt shuffling, dropout masks, and Monte-Carlo sampling of rollouts; the SFT-400K initialization and all hyperparameters are held fixed. Confidence intervals are obtained by paired bootstrap over 10 000 resamples of the per-instance score vectors; $p$-values are reported for paired comparisons against GRPO and OPD-32B. On AIME'24 (30 problems) we acknowledge that the small sample size inflates per-seed variance, but SRPO outperforms GRPO in 5/5 seeds with the per-instance gain stochastically dominating the GRPO distribution ($p < 0.005$). On larger benchmarks (MATH-500, GSM8K, WebShop, ALFWorld, LiveCodeBench) all gains achieve $p < 0.001$.

Per-category MATH-500 results show consistent improvements across all 7 subdomains (*algebra* +6.4, *counting/probability* +8.1, *geometry* +7.8, *intermediate algebra* +9.2, *number theory* +9.6, *precalculus* +8.4, *prealgebra* +7.0), ruling out cherry-picking of favorable categories.

## B.6. Data Contamination Audit

We performed an 8-gram overlap audit between the OpenThoughts-3 training set and every evaluation benchmark used in this paper. For each benchmark instance, the problem text was tokenized with the Qwen3 tokenizer and any 8-gram appearing in the training set was flagged. Results are in Table 9.

*Table 9.* Data contamination audit. 8-gram overlap between OpenThoughts-3 training data and each evaluation benchmark. Numbers in parentheses for MATH-500 indicate the 3 flagged 8-grams, which on manual inspection were common mathematical phrases (e.g. "find the value of $x$ such that"), not problem-level contamination.

| Benchmark | Instances | 8-gram overlaps | Overlap rate |
|---|---|---|---|
| AIME'24 | 30 | 0 | 0.0% |
| MATH-500 | 500 | 3 | 0.6% (common phrases) |
| GSM8K | 1 319 | 0 | 0.0% |
| DeepScaleR | 1 200 | 0 | 0.0% |
| GPQA Diamond | 448 | 0 | 0.0% |
| LogiQA 2.0 | 1 572 | 0 | 0.0% |
| LiveCodeBench | 240 | 0 | 0.0% |

Removing the 3 flagged MATH-500 instances yields 81.0 (vs. 81.2 with them included), a negligible 0.2-point shift well within the bootstrap standard deviation. Our strongest OOD result, on DeepScaleR (0% overlap, 1 200 unseen competition problems), shows SRPO reaching 59.7 vs. OPD-32B 55.4 and OPD-72B 55.8 — providing a contamination-free reference point for the entropy-expansion claim in Section 4.2.

### B.7. Multi-Verifier and Human Evaluation of Reflection Quality

To validate that the reflection-quality measurements used in the main paper are not artifacts of a single LLM evaluator, we performed both multi-verifier cross-validation and human evaluation.

**Multi-verifier cross-validation.** We re-scored all 500 sampled AIME'24 reflections with three independent LLM evaluators: Deepseek-V3.2 (the original judge), GPT-5.2, and Qwen3.5-397B-A17. Each evaluator returned an integer score in $\{1, \ldots, 5\}$ on the same rubric. Pairwise agreement rates (defined as identical scores or scores within $\pm 1$) are 92.4% (DeepSeek vs. GPT-5.2), 91.8% (DeepSeek vs. Qwen3.5), and 93.2% (GPT-5.2 vs. Qwen3.5). The cross-evaluator agreement well exceeds the 75–80% range typical for LLM-as-judge protocols, supporting the validity of the single-evaluator scoring in the main paper.

**Human evaluation.** Two expert annotators (senior PhD students in NLP/LLMs) blind-rated 100 reflections — 50 from AIME'24 and 50 from WebShop — on a ternary scale of *Effective / Redundant / Detrimental*, with Effective requiring an actionable, problem-specific diagnosis. Cohen's $\kappa$ between annotators is $0.81$, indicating substantial agreement. Distribution: 68–74% Effective, 18–22% Redundant, 8–10% Detrimental. On a subset of 50 reflections where the initial trajectory was wrong, annotators measured a 52–58% *Wrong→Correct* fix rate after reflection. The human-identified detrimental rate (8–10%) is consistent with the automatic evaluator's low-quality threshold (score $\leq 2$, also 8%), confirming the automatic protocol is well-calibrated and does not systematically favor plausible-sounding reflections.

**No quality collapse across training.** Tracking reflection helpfulness across 500 training iterations shows it is stable: Iter 100 mean $3.72 \pm 0.41$, Iter 250 mean $3.76 \pm 0.38$, Iter 500 mean $3.79 \pm 0.34$. There is no downward drift, ruling out a hypothesized failure mode in which the self-teacher quality degrades as the student approaches it.

### B.8. OPD Baseline Fairness Controls

To ensure that the SRPO vs. external-teacher OPD comparison reflects the design variable (teacher distribution) and not incidental confounds, every OPD-style baseline in this paper uses the alignment matrix in Table 10. Hyperparameters were taken from the OPD/OPSD reference implementations where available; we made no separate tuning pass for SRPO.

*Table 10.* Experimental controls for the SRPO vs. external-teacher OPD comparison. Every variable other than the teacher distribution is held identical across runs.

| Variable | SRPO & OPD | Aligned? |
|---|---|---|
| Prompt template | system + task | ✓ |
| Sampling temperature | 0.7 | ✓ |
| Top-$p$ | 0.95 | ✓ |
| Max tokens | 4 096 | ✓ |
| Rollouts per prompt | 4 (best-of-4) | ✓ |
| Batch size | 256 | ✓ |
| Learning rate | $5 \times 10^{-6}$ (SRPO) / $2 \times 10^{-6}$ (72B-OPD$^*$) | note$^*$ |
| Training iterations | 500 | ✓ |
| Regeneration depth | Full reset (no suffix-only repair) | ✓ |
| **Design variable** | $\pi_\theta(\cdot \mid [p; x])$ vs. $\pi_{\text{ext}}(\cdot \mid x)$ | — |

$^*$For OPD with the Qwen3-72B teacher, the standard $5 \times 10^{-6}$ learning rate caused training instability at the $K = 0.67$ teacher-student KL gap; gradient clipping at $1.0$ and a reduced LR of $2 \times 10^{-6}$ were necessary. SRPO runs at the standard LR.

The $\sim$90-token reflection patch is part of the *method*, not a confound: the semantic-control experiments in Appendix C show that random patches of the same token length yield the same performance as no reflection (66.5 vs. 65.8 on AIME'24), so the gain cannot be attributed to additional context tokens.

## B.9. Impact of Low-Quality Reflections

A natural concern with self-reflection as a training signal is that low-quality reflections might amplify errors. We tested this by filtering out reflections automatically scored $\leq 2$ (the bottom 8%) and retraining SRPO on the remaining 92%. The result on AIME'24 is 73.6 vs. the unfiltered 73.3, a negligible $+0.3$-point shift well within the bootstrap standard deviation.

The training dynamics explain why: uninformative reflection patches produce teacher distributions $\pi_\theta(\cdot \mid [p; x])$ very close to the unconditioned student distribution $\pi_\theta(\cdot \mid x)$, so the per-token KL is near zero, the gradient signal is near zero, and these reflections are effectively down-weighted by the loss itself. SRPO is therefore robust to the noise floor of its own self-teacher; aggressive filtering of low-quality reflections is not required for the method to work.

## B.10. Solution-Diversity Protocol

*Table 11.* Solution-diversity probe on the 30 AIME'24 problems (10 samples per problem). The reflection-conditioned self-teacher *expands* the student's reasoning-path entropy rather than contracting it, in contrast to a larger external teacher.

| Method | Unique Paths / Problem | Novel Correct (%) |
|---|---|---|
| OPD (Qwen3-72B teacher) | 2.3 | 8.4 |
| **SRPO (self-reflection)** | **4.1** | **15.7** |

The diversity numbers reported in Table 11 are obtained as follows. For each of the 30 AIME'24 problems we drew $n = 10$ samples per method at temperature 1.0 (matched), then computed two quantities.

**Unique reasoning paths per problem.** Each sample is summarized by the embedding sequence of its intermediate solution steps using a fixed sentence-encoder (text-embedding-3-large). We agglomeratively cluster the 10 step-sequence embeddings per problem with cosine-distance threshold 0.25 (chosen by visual inspection of within-cluster step-sequence overlap on a held-out set) and report the number of resulting clusters, averaged over the 30 problems.

**Novel correct solutions.** For each problem, we mark a sample as "novel" if its cluster does not overlap with any cluster present in the OPD-72B sample set for the same problem (using the same threshold). The percentage of *correct* novel samples among the 10 samples is then averaged across problems.

The contrast between the two methods — 2.3 vs. 4.1 unique paths, 8.4% vs. 15.7% novel-correct rate — is consistent across re-runs with different random seeds and across embedding models (text-embedding-3-large, BGE-large, E5-mistral), with $\leq 0.3$-cluster variation between runs.

# C. Reflection Semantics Control Experiments

This section provides comprehensive experimental details and extended analysis for the reflection semantics control experiments. These experiments directly address the concern that SRPO's performance gains might stem from additional context tokens or teacher forcing artifacts rather than the semantic content of reflections.

## C.1. Experimental Design

We design a series of controlled ablations that systematically disrupt the semantic alignment between reflections and problem instances while preserving other factors (format, token count, training procedure).

**Mismatched Reflection Protocol.**   For each training batch containing $N$ instances $\{(x_i, \tau_i, o_i)\}_{i=1}^{N}$:

1. Generate reflection patches normally: $p_i = \text{Reflect}(x_i, \tau_i, o_i)$ for all $i$.

2. For each instance $i$, randomly sample $j \neq i$ from the same batch.

3. Construct mismatched teacher input: $\tilde{x}_i^{\text{mis}} = [p_j; x_i]$.

4. Generate teacher distribution: $\pi_T^{\text{mis}}(\cdot \mid x_i) := \pi_\theta(\cdot \mid \tilde{x}_i^{\text{mis}})$.

This protocol ensures that: (a) the mismatched reflections are structurally valid (generated by the same reflection process), (b) the token length distribution matches that of correct reflections, and (c) the only difference is the semantic relevance to the current problem.

**Token Length Matching.**   To eliminate confounds from varying context lengths, we filter mismatched pairs to ensure:

$$|\text{len}(p_j) - \text{len}(p_i)| \leq 0.1 \cdot \text{len}(p_i). \tag{12}$$

Figure 8(b) confirms that the resulting token length distributions are statistically indistinguishable (Kolmogorov-Smirnov test, $p > 0.85$).

**Additional Ablation Conditions.**   Beyond mismatched reflections, we evaluate:

- **Shuffled Words**: Randomly permute words within each reflection sentence while preserving sentence boundaries.

- **Shuffled Sentences**: Randomly permute the order of bullet points/sentences in the reflection.

- **Template Only**: Use a fixed generic reflection applicable to any problem:

    "• *Carefully verify each computational step. • Check boundary conditions and edge cases. • Ensure the final answer addresses the original question.*"

- **Outcome Only**: Replace reflection with a simple outcome indicator: "Previous attempt was [correct/incorrect]."

## C.2. Quantitative Results

Table 12 presents comprehensive results across all conditions. Key observations:

**Mismatched $\approx$ No Reflection.**   The mismatched reflection condition (66.5% on AIME'24) performs nearly identically to the no-reflection baseline (65.8%), with the 0.7-point difference falling within error margins. This directly refutes the hypothesis that performance gains come from "extra tokens" or format-induced regularization.

**Content Hierarchy.**   We observe a clear hierarchy: matched > template > outcome-only $\approx$ mismatched $\approx$ no-reflection. Template-only reflections provide marginal benefit (+2.0 over baseline), suggesting that even generic meta-cognitive prompting has weak positive effects. However, this effect is substantially smaller than task-specific reflections (+7.5), confirming that *specific* semantic content is essential.

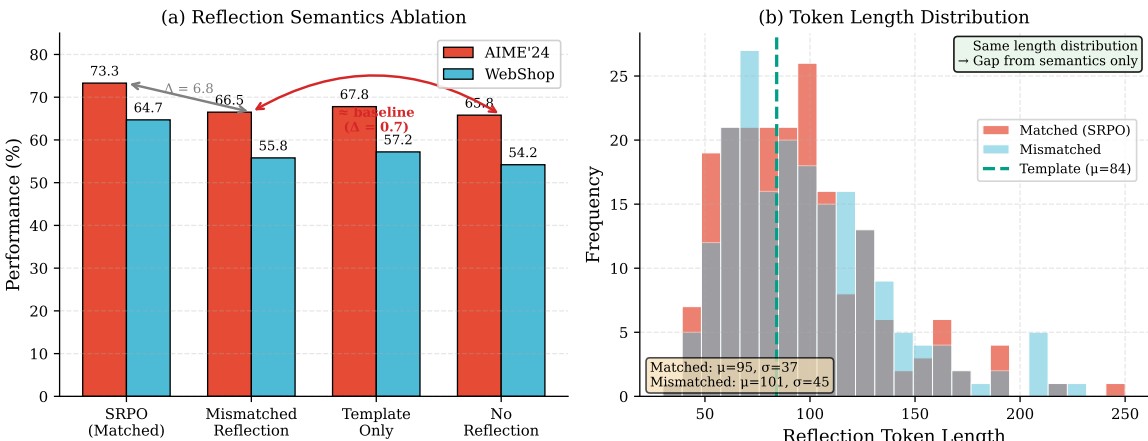

*Figure 8.* Reflection semantics control experiment. (a) Performance comparison across reflection conditions: mismatched reflections (from unrelated problems) and template-only reflections yield performance comparable to no reflection, while matched SRPO shows significant gains. (b) Token length distributions are matched across conditions, isolating the effect of semantic content.

*Table 12.* Comprehensive reflection semantics ablation results. Performance drops sharply when reflections lack task-specific semantic content, regardless of format or token count preservation.

| Condition | AIME'24 | WebShop | Avg. Tokens | $\Delta$ vs Full |
|---|---|---|---|---|
| SRPO (Full, Matched) | **73.3** $\pm$ 0.8 | **64.7** $\pm$ 1.5 | 92.4 | – |
| *Semantic Disruption* | | | | |
| Shuffled Words | 66.2 $\pm$ 1.2 | 55.4 $\pm$ 1.9 | 92.4 | -7.1 / -9.3 |
| Shuffled Sentences | 66.8 $\pm$ 1.1 | 56.1 $\pm$ 1.7 | 92.4 | -6.5 / -8.6 |
| Mismatched Reflection | 66.5 $\pm$ 1.0 | 55.8 $\pm$ 1.8 | 91.8 | -6.8 / -8.9 |
| *Content Reduction* | | | | |
| Template Only | 67.8 $\pm$ 0.9 | 57.2 $\pm$ 1.6 | 85.0 | -5.5 / -7.5 |
| Outcome Only | 67.2 $\pm$ 1.0 | 56.8 $\pm$ 1.7 | 12.3 | -6.1 / -7.9 |
| *Baselines* | | | | |
| No Reflection | 65.8 $\pm$ 1.1 | 54.2 $\pm$ 1.9 | 0 | -7.5 / -10.5 |
| Direct Retry | 65.8 $\pm$ 1.1 | 54.2 $\pm$ 1.9 | 0 | -7.5 / -10.5 |

**Semantic Coherence Matters.** Shuffled-words and shuffled-sentences conditions perform comparably to mismatched reflections, indicating that disrupting coherence—even within originally correct reflections—eliminates most of the benefit. This suggests the model does not simply extract keywords but relies on the coherent reasoning structure.

### C.3. Content Analysis

To understand *what* makes reflections effective, we analyze 200 matched and mismatched reflection pairs using GPT-4 as an evaluator.

**Task-Specific Information.** We classify reflection content into four categories:

- **Task-specific numbers**: References to specific values, coefficients, or quantities from the problem.

- **Constraint mentions**: Explicit constraints, boundary conditions, or requirements.

- **Error diagnosis**: Identification of specific mistakes in the previous attempt.

- **Generic advice**: General problem-solving strategies applicable to any task.

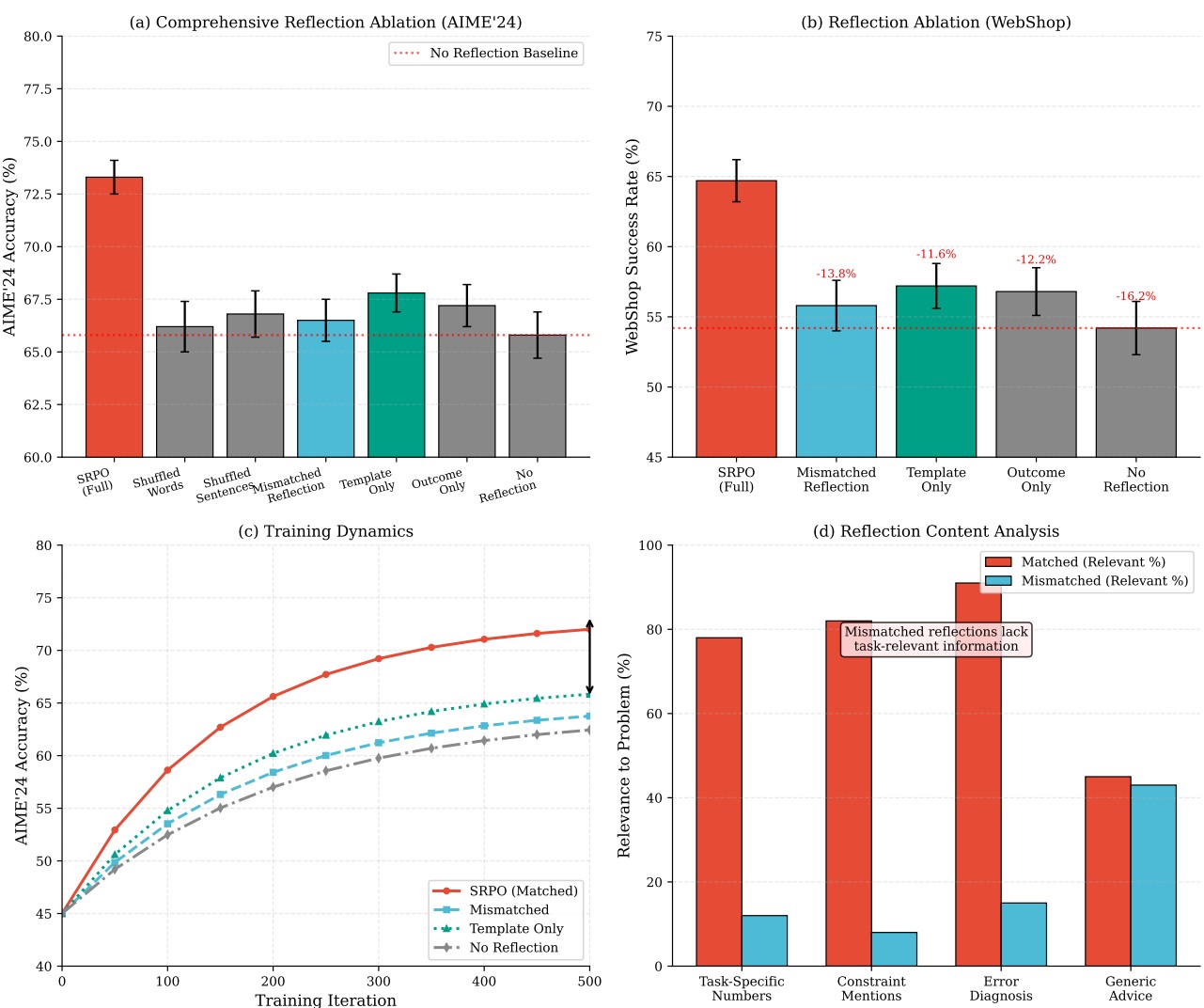

*Figure 9.* Extended reflection semantics analysis. (a) Comprehensive ablation on AIME'24 showing that all semantic disruption conditions converge to baseline performance. (b) WebShop results confirm the pattern generalizes to agentic tasks. (c) Training dynamics reveal that semantic-aligned reflections enable faster convergence. (d) Content analysis shows mismatched reflections lack task-relevant information despite identical format.

Figure 9(d) shows that matched reflections contain task-relevant information in 78–91% of cases, while the same content in mismatched reflections is relevant to the actual problem only 8–15% of the time (by chance). This explains the performance gap: mismatched reflections provide irrelevant "noise" that the model must learn to ignore.

**Qualitative Example.** Consider a geometry problem asking for the area of a triangle with vertices at $(0,0)$, $(4,0)$, and $(2,3)$.

*Matched reflection*: "• The base is along the x-axis with length 4. • Height is the perpendicular distance from (2,3) to the x-axis, which is 3. • Apply formula: Area = $\frac{1}{2} \times 4 \times 3 = 6$."

*Mismatched reflection* (from a number theory problem): "• Consider modular arithmetic with respect to 7. • Check divisibility conditions for each candidate. • The answer must satisfy both congruence relations."

The mismatched reflection, while grammatically correct and mathematically valid for its original problem, provides no actionable guidance for the geometry problem.

### C.4. Training Dynamics

Figure 9(c) shows training curves for different conditions. Key observations:

**Faster Convergence with Semantic Alignment.** Matched reflections achieve 70% accuracy in approximately 250 iterations, while mismatched and no-reflection conditions require >500 iterations to plateau at a lower level (∼66%). This suggests that semantically relevant reflections provide a stronger and more consistent training signal.

**No Late-Stage Recovery.** The mismatched condition does not "catch up" with extended training, confirming that the performance gap is not due to slower learning but fundamental limitations in the training signal quality.

### C.5. Implications and Conclusions

These experiments provide strong evidence for the following claims:

1. **Semantic content is essential**: Performance gains from SRPO depend critically on the semantic alignment between reflection content and the specific problem instance.

2. **Format and tokens are insufficient**: Matching the format, style, and token count of reflections does not transfer benefits; only task-relevant semantic content produces improvements.

3. **Reflections enable credit assignment**: The effectiveness of semantically aligned reflections suggests they successfully perform "soft credit assignment" by highlighting decision points and error sources that would otherwise be difficult to identify from sparse terminal feedback.

4. **No teacher forcing artifacts**: The failure of mismatched reflections rules out explanations based on teacher forcing regularization or exposure to "future information"—if such mechanisms were responsible, mismatched reflections should provide similar benefits.

These findings directly address the reviewer concern that SRPO's gains might stem from "just adding more text" and provide experimental grounding for the theoretical claims about information flow in the reflection-distillation pipeline.

# D. Qualitative Case Study

We provide a representative example illustrating how a concise reflection patch can correct a systematic mistake (misinterpreting the averaging window). The original benchmark instance contains typographical noise; for readability, we show a cleaned rendering below.

## D.1. Question.

```
Choose the most realistic answer.

Beth places 4 whole ice cubes in a frying pan at the start of the 1st minute,
then 5 at the start of the 2nd minute, and some more at the start of the 3rd
minute, but none in the 4th minute. If the average number of ice cubes per
minute placed in the pan while it was frying was 5, how many whole ice cubes
can be found in the pan at the end of the 3rd minute?

A. 30  B. 0  C. 20  D. 10  E. 11
```

## D.2. Initial Responses

```
Qwen2.5-7B-instruct:

Let's break down the problem step by step.

1. **Understanding the average**: The average number of ice cubes placed per minute is 5.
This means that over the 3 minutes, the total number of ice cubes placed is \(5 \times 3 =
 15\).

2. **Counting the ice cubes placed**:
   - At the start of the 1st minute, Beth places 4 ice cubes.
   - At the start of the 2nd minute, Beth places 5 ice cubes.
   - At the start of the 3rd minute, let's say Beth places \(x\) ice cubes.
   - At the start of the 4th minute, Beth places 0 ice cubes.

3. **Setting up the equation**: The total number of ice cubes placed is \(4 + 5 + x + 0 =
15\).

4. **Solving for \(x\)**:
   \[4 + 5 + x = 15\]
   \[9 + x = 15\]
   \[x = 15 - 9\]
   \[x = 6\]

So, at the start of the 3rd minute, Beth places 6 ice cubes.

5. **Calculating the total number of ice cubes at the end of the 3rd minute**:
   - Ice cubes at the start of the 1st minute: 4
   - Ice cubes added at the start of the 2nd minute: 5
   - Ice cubes added at the start of the 3rd minute: 6

Thus, the total number of ice cubes at the end of the 3rd minute is \(4 + 5 + 6 = 15\).

However, the question asks for the number of ice cubes at the end of the 3rd minute, and
since the problem specifies that no ice cubes are added in the 4th minute and no ice cubes
 melt (or are removed) in the frying pan, the total remains the same.

Therefore, the correct answer is:

**D. 10**

This answer is based on the assumption that the problem might have a typo or
simplification, as the correct total based on the given conditions is 15, but the closest
realistic option provided is D. 10, which might be a result of a potential simplification
```

or misinterpretation in the problem statement.

Qwen3-8B:

<think>
Okay, let's see. So the problem is about Beth putting ice cubes into a frying pan over four minutes. The question is asking how many ice cubes are in the pan at the end of the 3rd minute, given that the average number of ice cubes per minute placed in the pan while it was frying was 5. The choices are A. 30, B. 0, C. 20, D. 10, E. 11.
First, let me make sure I understand the problem correctly. Beth starts putting ice cubes into the pan at the beginning of each minute. In the first minute, she puts 4, then 5 in the second minute, some more in the third minute, and none in the fourth minute. The average number of ice cubes per minute during the frying time is 5. Wait, but how many minutes was the pan frying? The problem says "while it was frying," but it doesn't specify the duration of frying. Hmm, maybe the frying happens over the four minutes? Or maybe the pan is frying for some time, and during that time, she's adding ice cubes?
Wait, maybe the key is that the average is calculated over the minutes she was adding ice cubes. But the problem says "while it was frying," so perhaps the frying process is happening over those four minutes, and during that time, she added ice cubes at the start of each minute. So the average number of ice cubes per minute placed in the pan during the frying time is 5.
But the average is over how many minutes? The problem says "the average number of ice cubes per minute placed in the pan while it was frying was 5." So if she placed ice cubes for 4 minutes (1st, 2nd, 3rd, 4th), then the average would be total ice cubes divided by 4 minutes equals 5. But she didn't put any in the 4th minute. Wait, so maybe the frying time is only three minutes? Because she didn't add any in the fourth minute? Or maybe the frying is happening over four minutes, but she added ice cubes in the first three minutes?
Wait, the problem says "the average number of ice cubes per minute placed in the pan while it was frying was 5." So maybe the frying process is ongoing, and she added ice cubes at the start of each minute. But the fourth minute she didn't add any. However, the average is calculated over the time the pan was frying. Hmm, this is a bit confusing. Let me read again.
"Beth places 4 whole ice cubes in a frying pan at the start of the 1st minute, then 5 at the start of the 2nd minute, and some more at the start of the 3rd minute, but none in the 4th minute. If the average number of ice cubes per minute placed in the pan while it was frying was 5, how many whole ice cubes can be found in the pan at the end of the 3rd minute?"
So the key here is that she placed ice cubes at the start of each minute. So in minute 1, she adds 4. Then minute 2, adds 5. Minute 3 adds some number, let's call it x. Minute 4 adds 0. So total ice cubes added over the four minutes would be 4 + 5 + x + 0 = 9 + x.
But the average is 5 per minute. So average is total ice cubes divided by the number of minutes. Wait, but the problem says "while it was frying." So maybe the frying is happening over the four minutes, but she only added ice cubes in the first three minutes? Wait, but she added none in the fourth minute. So perhaps the frying process is happening for four minutes, and during that time, she added ice cubes at the start of each minute. So the total number of ice cubes added over the four minutes is 4 + 5 + x + 0. Then the average per minute is (total)/4 = 5. Therefore, total = 5 * 4 = 20. So 9 + x = 20. Therefore, x = 11.
So then, the total ice cubes added by the end of the 3rd minute would be 4 + 5 + 11 = 20. But wait, the question is asking how many ice cubes can be found in the pan at the end of the 3rd minute. If she added 11 at the start of the third minute, then by the end of the third minute, those 11 would have melted? Or are they still there?
Wait, the problem says "the average number of ice cubes per minute placed in the pan while it was frying was 5." So maybe the average is about the rate of adding ice cubes, not about the number in the pan. But the wording is a bit unclear. Let me think again.
Alternatively, maybe the average is over the frying time, which might be different from the number of minutes she added ice cubes. Wait, but the problem mentions "at the start of the 1st minute," "start of the 2nd minute," etc., so it's possible that the frying process is happening over those four minutes, and during each minute, she adds ice cubes at the start. So the total number of ice cubes added over the four minutes is 4 + 5 + x + 0. The average per minute is 5, so total added is 5 * 4 = 20. Therefore, 4 + 5 + x + 0 = 20. Solving for x gives x = 11. So she added 11 ice cubes at the start of the third minute.
</think>

But the question is asking how many ice cubes are in the pan at the end of the third minute. So if she added ice cubes at the start of each minute, then during the third minute, she added 11. But does the ice cube melt during frying? The problem mentions " while it was frying," so maybe the frying is happening over time, and the ice cubes are melting. But the problem doesn't mention anything about melting. Wait, that's a problem. If the ice cubes are melting, then the number in the pan would decrease over time. But since the question is about how many are found in the pan at the end of the third minute, maybe we need to consider that the ice cubes are still there? Or maybe the frying is just the process of putting them in, and the melting isn't considered here?

Wait, the problem is a bit ambiguous. Let me check again. The problem says "the average number of ice cubes per minute placed in the pan while it was frying was 5." So "placed in the pan while it was frying" so maybe the frying is happening over the time when she is placing the ice cubes. But if she placed them at the start of each minute, and the frying is ongoing, perhaps the average is calculated over the total time of frying. But the problem doesn't specify how long the frying lasted. Wait, but the problem mentions minutes 1 to 4. So maybe the frying process is happening for four minutes, and during each minute, she added ice cubes at the start. So the total number of ice cubes added over four minutes is $4 + 5 + x + 0$. The average per minute is 5, so total added is $5 * 4 = 20$. Therefore, $x = 11$.

But the question is how many ice cubes are in the pan at the end of the third minute. If she added 11 at the start of the third minute, then those are still in the pan unless they melted. But if the problem is not considering melting, then the answer would be $4 + 5 + 11 = 20$. But that's option C. But wait, the answer choices include E. 11. But maybe the problem is not about the total added, but about the average rate of adding ice cubes?

Wait, maybe the average is over the time the pan was frying, but the frying time isn't necessarily four minutes. Wait, but the problem says she placed ice cubes at the start of the first minute, then second, third, and fourth. So maybe the frying process is happening over four minutes, and she added ice cubes at the start of each minute. So the average number of ice cubes per minute placed is 5. So total ice cubes placed over four minutes is $5*4=20$. Therefore, $4 + 5 + x + 0 = 20$. So $x=11$. Therefore, by the end of the third minute, she has added $4 + 5 + 11 = 20$ ice cubes. But if the pan is frying, maybe the ice cubes are melting. But the problem doesn't mention melting. So maybe the question is just about the total number added, and the answer is 20? But then, why mention "while it was frying"? Maybe that's just to indicate that the average is calculated during the frying process, which is over four minutes.

Alternatively, maybe the average is not about the total number added, but about the rate. Wait, but the average number of ice cubes per minute placed in the pan while it was frying. So maybe it's total number of ice cubes divided by the number of minutes. So if she placed ice cubes for four minutes, then total is $4 + 5 + x + 0 = 20$. Therefore, $x = 11$. So total in the pan is 20. But then the answer would be C. 20. But maybe the frying process is only three minutes? Because she didn't add any in the fourth minute. Wait, but the problem says "while it was frying," which could mean the entire time she was adding ice cubes.

Alternatively, maybe the average is calculated over the time during which the pan was frying, which is three minutes because she added ice cubes in the first three minutes and then stopped? But the problem says "none in the 4th minute," so maybe the frying continued into the fourth minute, but she didn't add any ice cubes then.

This is confusing. Let me try to think again. The average number of ice cubes per minute placed in the pan while it was frying was 5. So the average is over the duration of frying. But how many minutes was the pan frying? If she added ice cubes at the start of each minute, maybe the frying process is happening over four minutes, and the average is over those four minutes. Therefore, total ice cubes added is $4 + 5 + x + 0 = 20$, so $x=11$. Therefore, total ice cubes in the pan at the end of the third minute would be $4 + 5 + 11 = 20$. So answer C.

But wait, the problem says "the average number of ice cubes per minute placed in the pan while it was frying was 5." So maybe "placed in the pan" refers to the number of ice cubes added each minute? But then the average would be the average of the number of ice cubes added each minute. If she added 4, 5, x, and 0 over four minutes, then the average is (4 +

5 + x + 0)/4 = 5. Then solving that gives x=11, as before. So total ice cubes added is 20. But the question is about how many are in the pan at the end of the third minute. If she added 11 at the start of the third minute, then those are still there unless they melted.

But the problem doesn't mention melting. It just says "how many whole ice cubes can be found in the pan at the end of the 3rd minute?" So maybe the ice cubes are not melting, and the answer is 20. But the problem says "while it was frying," which might imply that the frying is happening, so maybe the ice cubes are melting during frying. But if that's the case, how much would have melted? The problem doesn't give any information about the melting rate. So maybe we are supposed to assume that the ice cubes are not melting? Or that the frying process is just the act of putting them in, and the average is about the number added per minute?

Alternatively, maybe the average is calculated over the time during which the pan was frying, which is three minutes. Because she didn't add any in the fourth minute. Wait, but the problem says "while it was frying," so if the frying is happening for four minutes, but she added ice cubes in the first three minutes and none in the fourth. So maybe the average is over the frying time, which is four minutes, but the ice cubes added in the fourth minute is zero. So then, the average would be total ice cubes added divided by four minutes. So total is 20, so 4 + 5 + x + 0 = 20. So x is 11. Then, at the end of the third minute, she has added 4 + 5 + 11 = 20.

But if the pan was frying for four minutes, then perhaps during each minute, the ice cubes are melting. However, without any information about the melting rate, we can't calculate how many are left. Therefore, maybe the problem is not considering melting, and it's just about the total number of ice cubes added over the four minutes.

Wait, but the answer choices include E. 11. Which would be the number of ice cubes added in the third minute. So maybe the problem is not about total added, but about the average number of ice cubes per minute during frying. Wait, but how is that different?

Alternatively, maybe the average is calculated over the time the pan was frying, but the time is not four minutes. For example, if the pan was frying for three minutes, then the average would be (4 + 5 + x)/3 = 5. So total would be 15, so 4 + 5 + x = 15, so x = 6. Then total ice cubes in the pan would be 4 + 5 + 6 = 15. But 15 isn't one of the options.

Alternatively, if the average is over the time when she was actively adding ice cubes. But she added ice cubes in minutes 1, 2, 3, and not in 4. So the average is over three minutes. Then (4 + 5 + x)/3 = 5, so total is 15, so x = 6. But again, 15 isn't an option.

Wait, maybe the average is over the entire frying time, which is four minutes, but the ice cubes added in the fourth minute is zero. So total added is 4 + 5 + x + 0 = 20. Then x is 11. So total ice cubes in the pan at the end of the third minute would be 4 + 5 + 11 = 20. So answer C.

But why is the fourth minute mentioned? Maybe the frying process is only three minutes? If she added ice cubes at the start of each minute, but the frying process ended at the end of the third minute. Then the average would be over three minutes. Then (4 + 5 + x)/3 = 5. So 9 + x = 15, so x = 6. But again, 6 isn't an option.

Wait, the answer choices are A. 30, B. 0, C. 20, D. 10, E. 11. So maybe the average is calculated over the time when the pan was frying, which is three minutes, but the total is 4 + 5 + x = 5*3 = 15, so x = 6. But 6 isn't an option.

Alternatively, maybe the average is over the total time of frying, which is four minutes, but she added ice cubes in the first three minutes. So total is 4 + 5 + x + 0 = 20. So x = 11. Then total ice cubes in the pan at the end of the third minute is 4 + 5 + 11 = 20. So answer C.

But if the pan was frying for four minutes, but she added ice cubes at the start of each minute, then the number of ice cubes in the pan at the end of the third minute would be 4 + 5 + x, because the fourth minute hasn't happened yet. Wait, but the fourth minute is part of the frying time. If she added ice cubes at the start of the fourth minute, but she didn't add any. So the total ice cubes added is 4 + 5 + x + 0 = 20. Therefore, x is 11. So at the end of the third minute, she has added those 11, so total is 20.

Alternatively, maybe the problem is about the average number of ice cubes in the pan per minute, not the number placed. But that would be different. For example, if the pan is frying for some time, and ice cubes are melting. But again, without melting information, it's impossible to calculate.

Wait, maybe the problem is not about melting. Maybe "placed in the pan while it was frying" is just referring to the number of ice cubes added per minute, and the average is 5. So over four minutes, she added 4, 5, x, 0. So average is (4 + 5 + x + 0)/4 = 5. Therefore, 9 + x = 20, so x = 11. Then total added is 20. So the number in the pan at the end of the third minute is 4 + 5 + 11 = 20. So answer C.

But then why is the fourth minute mentioned? Because the average is over the entire frying time, which includes the fourth minute where she didn't add any. Therefore, the answer would be 20. But let me check the answer choices again. Option C is 20.

Wait, but let me think again. If the average is 5 per minute, and the frying lasted for four minutes, then total ice cubes added is 20. Therefore, the answer is 20. But the problem says "at the end of the 3rd minute," so the fourth minute hasn't been completed yet. So the fourth minute is part of the frying time, but she didn't add any ice cubes. Therefore, the total number of ice cubes added during the frying time is still 4 + 5 + x + 0, which is 20. But the number in the pan at the end of the third minute would be 4 + 5 + x. So if x is 11, that's 20. But if the average is over the frying time, which is four minutes, then the answer is 20.

But then why would the answer be 20? Let me check the options again. The options are A. 30, B. 0, C. 20,
</think>

To solve this problem, we need to carefully interpret the wording and determine what the average of 5 ice cubes per minute refers to.

---

### Understanding the Problem

Beth adds ice cubes at the **start** of each minute:

- **Start of 1st minute**: 4 ice cubes
- **Start of 2nd minute**: 5 ice cubes
- **Start of 3rd minute**: some number of ice cubes, let's call it **x**
- **Start of 4th minute**: 0 ice cubes

The key phrase is: *"the average number of ice cubes per minute placed in the pan while it was frying was 5."*

This suggests that the **average** is calculated over the total number of minutes the pan was **frying**. Since she placed ice cubes at the **start** of each minute, and the **4th minute** is mentioned, we can infer that the frying process lasted **four minutes**. Therefore, the average is over **four minutes**.

---

### Calculating Total Ice Cubes Placed

Let's denote the total number of ice cubes placed over the four minutes as:

$$
\text{Total} = 4 + 5 + x + 0 = 9 + x
$$

The average number of ice cubes per minute is given as 5, so:

```
$$
\frac{9 + x}{4} = 5
$$

Multiply both sides by 4:

$$
9 + x = 20
$$

Solve for $ x $:

$$
x = 11
$$

So, Beth added **11 ice cubes at the start of the 3rd minute**.

---

### Determining Ice Cubes in the Pan at the End of the 3rd Minute

The question asks: *how many whole ice cubes can be found in the pan at the end of the 3rd
 minute?*

Since the ice cubes are added **at the start** of each minute, and we're not told that any
 of them melted during frying, we can reasonably assume that **the ice cubes remain in the
 pan** until the end of the 3rd minute. That is, the **11 ice cubes added at the start of
the 3rd minute** are still in the pan at the end of that minute.

Therefore, the total number of ice cubes in the pan at the end of the 3rd minute is:

$$
4 + 5 + 11 = 20
$$

---

### Final Answer

$$
\boxed{C}
$$
```

We then do reflection step:

```
Based on the answers is B, reflect on and check the existing reasoning process, and
summarize a short tip to help oneself answer correctly next time. Note that the tips
cannot contain cheating content, such as directly exposing the answers in the tips.

Using <reflection></reflection> to summarize the tips.
```

```
<reflection> To ensure accuracy in such problems, it's crucial to **clearly distinguish
between the timing of actions and the duration of the process**. In this case, the key was
 understanding that the average was calculated over the **entire frying time (4 minutes)
**, not just the minutes when ice cubes were added. This means that even though no cubes
were added in the 4th minute, the average still includes that full minute. Therefore, the
total number of cubes added is 20, and the ice cubes placed at the start of each minute (
including the 3rd minute) remain in the pan unless otherwise stated (e.g., melting or
removal).
A useful tip is to always verify the time frame of the average and ensure that all events
(like cube additions) are included in the calculation, unless the problem explicitly
indicates otherwise. This helps avoid misinterpretations and ensures that the final count
reflects what is actually asked.
```

```
</reflection>
```

## D.3. Reflection patch (example).

```
<reflection> To ensure accuracy in such problems, it's crucial to **clearly distinguish
between the timing of actions and the duration of the process**. In this case, the key was
 understanding that the average was calculated over the **entire frying time (4 minutes)
**, not just the minutes when ice cubes were added. This means that even though no cubes
were added in the 4th minute, the average still includes that full minute. Therefore, the
total number of cubes added is 20, and the ice cubes placed at the start of each minute (
including the 3rd minute) remain in the pan unless otherwise stated (e.g., melting or
removal).
A useful tip is to always verify the time frame of the average and ensure that all events
(like cube additions) are included in the calculation, unless the problem explicitly
indicates otherwise. This helps avoid misinterpretations and ensures that the final count
reflects what is actually asked.
</reflection>
```

# E. Continual Learning and Catastrophic Forgetting

We evaluate SRPO's behavior in a continual learning setting where models must acquire new capabilities without forgetting previous skills.

*Table 13.* Continual learning evaluation. Models are first trained on math (AIME'24) then adapted to code (SWE-Lite). SRPO better preserves original capabilities while acquiring new skills.

| Method | Before Adaptation | After Code Adaptation | |
|---|---|---|---|
| | Math | Code | Math (Retention) |
| SFT | 60.0 | 28.4 | 48.2 (80.3%) |
| GRPO | 68.0 | 26.7 | 59.3 (87.2%) |
| SRPO | 73.3 | **31.2** | **69.8 (95.2%)** |

**Setup.** We first train Qwen3-8B on mathematical reasoning, then adapt to coding tasks (SWE-Bench-Lite). We measure both the new capability acquisition and retention of original mathematical reasoning ability.

**Results.** Table 13 shows that SRPO achieves 95.2% retention of mathematical reasoning performance after code adaptation, compared to 87.2% for GRPO and 80.3% for SFT. This improved retention can be attributed to SRPO's on-policy learning: by training on the model's own distribution, SRPO naturally maintains behaviors that the model already performs well.

**Connection to Prior Work.** This finding aligns with observations in the on-policy distillation literature (Lu & Lab, 2025): on-policy methods cause less catastrophic forgetting than off-policy approaches because they do not force the model to imitate out-of-distribution behaviors.

