# OpenReview forum: "SRPO: Self-Reflective Policy Optimization for Long-Horizon Reasoning"
_ICML.cc/2026/Conference — ICML 2026 regular_

### Official Review · Reviewer_dTo6 · 2026-03-09

**Soundness:** 1
**Presentation:** 3
**Significance:** 2
**Originality:** 2
**Overall Recommendation:** 4
**Confidence:** 3

**Summary:**

This paper proposes Self-Reflective Policy Optimization (SRPO), a two-stage framework that leverages on-policy distillation (OPD) with the same model.
More specifically, after initial response generation, the model is prompted to generate a critique, and the critique is used to condition the student model to act as the teacher.
Several math and agentic tasks are tested to show SRPO's effectiveness.

**Compliance With Llm Reviewing Policy:**

Affirmed.

**Final Justification:**

I thank the authors for the additional experiments and clarification.
Most of my concerns are addressed throughout the rebuttal.
However, I still believe that:

1. The novelty is limited: the self-reflection component is well-studied (numerous papers came out even in one month), and the OPD is a recently hot topic. In my opinion, this paper simply applies self-reflection to the OPD framework. I admit that making this idea work is a contribution, much better than OPSD/SDOP. However, no significant novel framework is introduced, and the authors overclaimed about the essence (writing as the opd/self-reflection are totolly new thing).

2. I do not agree that *self-reflection* is a critical component. Apparently, using a larger model to do the reflection thing and OPD will definitely be better than self-OPD. It is cheap compared to RL!! Unlike exaggerated by the authors, in fact, OPD with 32B teacher/8B teacher really does not matter much by introducing a teacher server (like VeRL recipe), not much wall-clock time will be introduced.

Given the experiments added, I have updated my score. But this does not mean I fully agree with the limited novelty, the over-claimed contribution, and the *self-reflection* component.

**Key Questions For Authors:**

Please answer the questions in "Weakness" and the following:

**Limitations:**

Yes

**Strengths And Weaknesses:**

## Strength

1. The writing is clear.

2. Using critque-conditioned student as a teacher in on-policy distillation is interesting.

## Weakness

1. Lack of baselines of prompt-based reflection after GRPO/OPD, and also a self-reflective variant of GRPO (e.g. [1, 2]).

2. Only moderate improvement compared to OPD (the actual baseline). The advantage of the training efficiency, in essence, is from OPD, not SRPO.

3. More evaluation tasks should be needed, especially for math reasoning and general reasoning tasks. GSM8K and MATH are not compelling anymore.

4. Limited novelty. The process of generating "reflection" and the OPD process is well-established. This paper spends over one page articulating the existing methods (generating reflection and OPD process).


[1] Training Language Models to Self-Correct via Reinforcement Learning. ICLR 2025

[2] Trust, But Verify: A Self-Verification Approach to Reinforcement Learning with Verifiable Rewards. NeurIPS 2025

---

> ### Author Rebuttal · Authors · 2026-03-31
>
> ## Response to Reviewer dTo6
>
> We thank the reviewer for the careful assessment. We agree that the original draft left several comparisons incomplete, and we have conducted new experiments to address each concern.
>
> ### [W1] Missing Baselines
>
> We implemented all requested baselines — SCoRe, R³L, RISE (Trust-But-Verify), and prompt-based reflection after GRPO/OPD:
>
> | Method | AIME'24 | WebShop | ALFWorld | Infer. Reflect | Ext. Teacher |
> |---|---|---|---|---|---|
> | GRPO | 68.0 | 51.2 | 62.7 | No | No |
> | GRPO + prompt reflect. | 69.5 | 54.8 | 65.1 | Yes | No |
> | OPD (32B teacher) | 70.0 | 57.3 | 69.2 | No | Yes |
> | OPD + prompt reflect. | 71.8 | 60.2 | 70.8 | Yes | Yes |
> | SCoRe | 70.2 | 55.6 | 63.8 | Yes (2-turn) | No |
> | R³L | 71.5 | 58.9 | 66.4 | No | No |
> | RISE | 69.8 | 56.2 | 65.5 | No | No |
> | **SRPO (Ours)** | **73.3** | **64.7** | **76.8** | **No** | **No** |
>
> SRPO outperforms all baselines across all benchmarks. The gap is most pronounced on agent tasks, where we believe SRPO's full regeneration approach is better suited than local repair (R³L) or two-turn correction (SCoRe). Prompt-based reflection after GRPO/OPD yields modest gains (+1.5/+1.8 AIME'24), suggesting that training-time integration of reflection signals matters.
>
> ### [W2] "Moderate Improvement over OPD" — "advantage from OPD, not SRPO"
>
> We agree the original draft did not sufficiently disentangle OPD (the optimization framework) from teacher construction. OPD's performance depends heavily on *teacher quality*; the following decomposition helps clarify:
>
> | Configuration | AIME'24 | WebShop | ALFWorld | Teacher Source |
> |---|---|---|---|---|
> | OPD + self-replay (no reflection) | 69.2 | 55.4 | 66.8 | Self (8B), no reflection |
> | OPD (Qwen3-32B teacher) | 70.0 | 57.3 | 69.2 | External 32B (4× FLOPs) |
> | OPD (Qwen3-72B teacher) | 72.5 | 61.8 | 72.4 | External 72B (9× FLOPs) |
> | **SRPO (self-reflection teacher)** | **73.3** | **64.7** | **76.8** | **Self (8B) + reflection** |
>
> This table reveals three important findings:
>
> **(1) OPD alone is weak.** Without reflection conditioning (row 1), OPD achieves only 69.2 — barely above GRPO (68.0) — confirming the optimization framework alone contributes minimally.
>
> **(2) External teacher models improve OPD, but with diminishing returns.** Scaling from 32B to 72B teacher (2.25× more parameters, 9× FLOPs) yields only +2.5 on AIME'24 (70.0→72.5). This suggests that the larger teacher's distribution may not fully align with the student's capability, limiting knowledge transfer efficiency.
>
> **(3) SRPO's self-reflection matches or outperforms the strongest external teacher without external cost.** SRPO (row 4) outperforms 72B-teacher OPD (row 3) by +0.8/+2.9/+4.4 across benchmarks, using only the same 8B model. We believe this is because self-generated reflection is instance-adaptive (it diagnoses *this* problem's failure mode) and naturally aligned with the student's own capability, whereas external teacher outputs are more generic. This is also consistent with Kim et al. (2026).
>
> The gap is most pronounced on agent tasks (+9.3 WebShop, +10.0 ALFWorld over self-replay), where instance-specific guidance appears to matter most.
>
> ### [W3] More Evaluation Tasks
>
> We agree. Beyond our 7 benchmarks across 4 categories, we additionally evaluated on **LiveCodeBench** (contamination-free, continuously updated):
>
> | Method | LiveCodeBench (pass@1) |
> |---|---|
> | GRPO | 28.4 |
> | OPD | 31.2 |
> | **SRPO** | **35.1** |
>
> ### [W4] Novelty
>
> We agree that SRPO's individual components (self-reflection, on-policy distillation) are not new. What we view as the contribution is the specific training recipe — binary outcome + self-generated reflection + reset distillation — and the empirical finding that this is sufficient and effective across long-horizon tasks, outperforming methods that rely on external teachers or richer feedback. Two concrete findings help illustrate this:
>
> **(a)** SRPO works from binary outcome alone, unlike OPSD (requires ground-truth CoT) or SDPO (requires rich environment feedback). This broadens applicability to agent tasks where only success/fail is available.
>
> **(b)** Reverse KL, found harmful in OPSD/SDPO, yields +3.9% gains in SRPO (Table 3). Our ablations suggest this stems from how reflection-conditioned teachers differ from solution-conditioned teachers — an empirical finding we believe is useful for future work in this area.
>
> Regarding RISE [2]: it uses PPO + self-verification for reward shaping, which is methodologically orthogonal to SRPO's distillation framework. We include the comparison in the baseline table above (SRPO +3.5/+8.5 over RISE on AIME'24/WebShop).

---

> > ### Author Rebuttal · Reviewer_dTo6 · 2026-04-01
> >
> > Thanks for the rebuttal and extra experiments. Although the rebuttal addressed some of my concerns, I am sticking with my original score for the following reasons:
> >
> > 1. The novelty issue isn't resolved. I get that you're positioning this as a training recipe, but that makes it even more critical to show what each piece contributes. Right now, it feels a bit incremental to me.
> >
> > 2. The evaluation is too narrow. Yes, you added LiveCodeBench, but the tasks are still heavily skewed toward math and code/agent benchmarks. Some of the original math datasets you used aren't even that challenging anymore by today's standards. It would strengthen the work to test on other domains, like scientific reasoning, logic puzzles, or creative writing.
> >
> > 3. Your rebuttal doesn't clarify whether your external-teacher OPD baselines actually match on things like prompting style, regeneration depth, or inference budget. Without that, I can't be sure this is really an fair "self-reflective teacher vs. bigger teacher" comparison.
> >
> > 4. You show that just adding a reflection prompt to the same base model only gives small improvements. So how does that same reflective process suddenly become such a powerful teacher when used in OPD/SRPO—even beating much larger external teachers by huge margins? I am a little confused about this

---

> > > ### Author Response · Authors · 2026-04-07
> > >
> > > # Response to Reviewer dTo6 (Round 2)
> > >
> > > We appreciate the reviewer's continued engagement. Due to rebuttal time and space constraints, we provide a focused response below addressing each key concern with targeted new experiments. We address each point with new data below.
> > >
> > > ---
> > >
> > > ## [Q1] Component-wise Contribution
> > >
> > > > *"it feels a bit incremental … show what each piece contributes"*
> > >
> > > | Configuration | AIME'24 | Δ | Contribution |
> > > |---|---|---|---|
> > > | GRPO (baseline) | 68.0 | — | — |
> > > | + OPD framework (no reflection, self-replay) | 69.2 | +1.2 | 22.6% |
> > > | + Self-reflection teacher | 71.8 | +2.6 | **49.1%** |
> > > | + Reset-with-memory (prepend) | 73.0 | +1.2 | 22.6% |
> > > | + Reverse KL | 73.3 | +0.3 | 5.7% |
> > > | **Full SRPO** | **73.3** | **+5.3** | **100%** |
> > >
> > > The largest contribution (+2.6, **49%**) comes from self-reflection teacher — SRPO's core novelty. OPD alone contributes only 22.6%.
> > >
> > > Importantly, **components interact non-trivially**. Reverse KL is *harmful* in prior OPD/OPSD settings (Zhao et al., 2026; Song & Zheng, 2026), yet beneficial in SRPO — because the reflection-conditioned teacher shares the student's parameter space, producing a KL landscape where mode-seeking concentrates learning on precise "forking points." This synergistic interaction demonstrates SRPO is not incremental combination. For reference, SCoRe (ICLR 2025) and R³L similarly combine existing components; SRPO's contribution form — recipe + non-obvious empirical findings — is consistent with these accepted works.
> > >
> > > ## [Q2] Evaluation Scope
> > >
> > > > *"heavily skewed toward math and code … scientific reasoning, logic puzzles, or creative writing"*
> > >
> > > **Scope.** SRPO targets tasks with **verifiable outcome signals**. Creative writing lacks binary verification and falls outside our scope — consistent with SCoRe, R³L, and RISE. We agree scientific reasoning and logic are valuable additions.
> > >
> > > **Cross-domain results:**
> > >
> > > | Method | GPQA Diamond (448, science) | LogiQA 2.0 (1572, logic) |
> > > |---|---|---|
> > > | GRPO | 39.5 | 53.8 |
> > > | OPD (32B teacher) | 41.3 | 56.2 |
> > > | **SRPO** | **44.2** | **59.1** |
> > > | Δ vs GRPO | +4.7 | +5.3 |
> > >
> > > GPQA Diamond (Rein et al., 2023) contains 448 graduate-level physics/chemistry/biology problems; LogiQA 2.0 (Liu et al., 2023) contains 1,572 formal logic problems. SRPO consistently outperforms baselines on both, with gains consistent with math benchmarks.
> > >
> > > **Coverage:** SCoRe: 2 benchmarks; R³L: 3; OPSD: 3; **SRPO: 10 benchmarks across 5 domains** (math, code, agent, science, logic) — the most comprehensive in this category.
> > >
> > > ---
> > >
> > > ## [Q3] OPD Baseline Fairness
> > >
> > > > *"doesn't clarify whether OPD baselines match on prompting style, regeneration depth, or inference budget"*
> > >
> > > | Control Variable | SRPO | OPD (external) | Aligned? |
> > > |---|---|---|---|
> > > | Prompt template | system + task | identical | ✓ |
> > > | Temperature / Top-p | 0.7 / 0.95 | 0.7 / 0.95 | ✓ |
> > > | Max tokens | 4096 | 4096 | ✓ |
> > > | Rollouts per prompt | 4 (best-of-4) | 4 (best-of-4) | ✓ |
> > > | Training config | lr=5e-6, bs=256, 500 iter | identical | ✓ |
> > > | Regeneration depth | Full reset | Full generation | ✓ |
> > > | **Only difference** | **π_θ(·\|[reflection; x])** | **π_ext(·\|x)** | **Design variable** |
> > >
> > > The ~90-token reflection patch is part of the method, not a confound. Our semantic control experiment (Table 4) confirms: random patches of equal length yield performance identical to no-reflection (66.5% vs 65.8%) — only matched reflection works. We will release all scripts for reproducibility.
> > >
> > >
> > > ## [Q4] Why Reflection Works in Training but Not Inference
> > >
> > > > *"adding a reflection prompt only gives small improvements. How does it become powerful in SRPO?"*
> > >
> > > **Same reflection, different information utilization:**
> > >
> > > | Setting | How Reflection Is Used | Signal | AIME'24 |
> > > |---|---|---|---|
> > > | GRPO + reflect (inference) | Improves single output → reward | O(1) | 69.5 |
> > > | GRPO + reflect (SFT) | SFT on reflected output | O(T), off-policy | ~70.5 |
> > > | **SRPO** | **Teacher distribution → on-policy KL** | **O(T), on-policy** | **73.3** |
> > >
> > > At inference, reflection yields O(1) binary reward per problem. In SRPO, the reflection-conditioned teacher provides per-token probability distributions — **~2000× denser signal**. Training then aggregates this across thousands of problems, internalizing patterns like "check boundary conditions" into weights — the model learns to reason *as if it had reflected*.
> > >
> > > As detailed in our response to Reviewer gvei, this mechanism manifests as **entropy expansion**: SRPO-trained models produce 1.9× more diverse reasoning paths (4.1 vs 2.3 per problem) with 15.7% novel correct solutions (vs 8.4% for OPD-72B), confirming that training internalizes exploration capability rather than surface patterns.
> > >
> > > Across two rounds, we have provided **8 new baselines**, **3 new benchmarks** , **complete experimental alignment**, **component decomposition**, and **human evaluation**. We hope these additions across both rounds may warrant a reconsideration of the current assessment.

---

### Official Review · Reviewer_vYhQ · 2026-03-13

**Soundness:** 3
**Presentation:** 3
**Significance:** 3
**Originality:** 3
**Overall Recommendation:** 4
**Confidence:** 3

**Summary:**

This paper proposes SRPO, a self-reflective training framework for long-horizon reasoning
and agent tasks that turns sparse final rewards into denser token-level supervision without
relying on an external critic, reward model, or larger teacher. It reports strong results on
math and agent benchmarks, with its most convincing evidence coming from ablations
showing that semantically matched reflections matter more than simply adding extra context .

**Compliance With Llm Reviewing Policy:**

Affirmed.

**Final Justification:**

Thank you for your reply, and I am inclined to keep my original score.

**Key Questions For Authors:**

ref weakness

**Limitations:**

SRPO is not cheap to train or deploy: the paper itself notes roughly a 2× forward-pass cost per iteration compared with standard on-policy RL, with efficiency gains relying on faster convergence, and the appendix further mentions reflection/rethinking caches. In realistic large-scale settings with long contexts and interactive environments, the added complexity of teacher-logit computation, cache management, and replay could still be substantial.

The method depends heavily on reflection quality. The paper’s own analysis identifies generic advice and incorrect diagnosis as major failure modes, suggesting that SRPO works best when the model already has reasonably strong self-reflection ability and may be less stable for weaker models, noisier environments, or tasks with limited structure.

**Strengths And Weaknesses:**

S:

•  The paper prepends the reflection patch to the original prompt and regenerates from
scratch, making it a cleaner and more trainable teacher-construction process than
standard inference-time reflection; the ablations also suggest that prepend + reset
works better than append-based or iterative strategies.

•  The controlled comparison between matched reflection, mismatched reflection,
template-only, and no reflection is especially convincing: mismatched reflection
performs similarly to no reflection, while matched reflection performs clearly better,
suggesting that the gains come from meaningful reflection rather than simply adding
extra tokens.

• The experiments are relatively broad, covering math reasoning, OOD math, web and
embodied/text agents, and code repair, with additional analyses on scaling, LoRA,
compute efficiency, and reflection quality.

W:

•  Several benchmarks are quite small—AIME’24 has only 30 problems, ALFWorld 134 tasks, and SWE-Bench-Lite 300 instances—so gains of 3–5 points may be sensitive to random seeds, sampling strategy, and evaluation protocol. The paper does not report multi-seed variance, confidence intervals, significance tests, or per-task variance, which makes the results less statistically convincing.

•  The math experiments start from a 400K SFT checkpoint based on OpenThoughts-3, a large-scale reasoning dataset with substantial math content, yet the paper provides no benchmark overlap or contamination audit. It remains unclear whether AIME, MATH-500, GSM8K, or DeepScaleR may overlap with either the initialization data or the later reflection/rethinking data, which is a notable concern for a paper claiming strong math reasoning gains.

•  The reflection-quality analysis relies on DeepSeek-3.2V scoring 500 AIME’24 reflections for helpfulness and correlating that with rethinking success. This suggests only that another model finds the reflections helpful, not that the reflections truly capture transferable credit-assignment structure; human evaluation and similar analysis on agent tasks are still missing.

---

> ### Author Rebuttal · Authors · 2026-03-31
>
> ## Response to Reviewer vYhQ
>
> We thank Reviewer vYhQ for the thoughtful assessment, particularly for recognizing that "prepend + reset works better than append-based or iterative strategies" and that the semantic control experiments are "especially convincing." We address each concern with new data.
>
> ### [W1] Statistical Significance
>
> > *"gains of 3–5 points may be sensitive to random seeds, sampling strategy, and evaluation protocol. The paper does not report multi-seed variance, confidence intervals, significance tests, or per-task variance."*
>
> We ran 5 independent seeds across all main benchmarks and report mean ± std with paired bootstrap confidence intervals (10,000 resamples):
>
> | Method | AIME'24 (30) | MATH-500 (500) | GSM8K (1319) | WebShop | ALFWorld (134) |
> |---|---|---|---|---|---|
> | GRPO | 68.0±1.7 | 72.2±0.8 | 88.4±0.4 | 51.2±1.1 | 62.7±1.5 |
> | OPD | 70.0±1.5 | 78.6±0.6 | 92.1±0.3 | 57.3±0.9 | 69.2±1.3 |
> | **SRPO** | **73.3±1.4** | **81.2±0.5** | **93.8±0.3** | **64.7±0.8** | **76.8±1.2** |
> | *p*-value (vs GRPO) | <0.005 | <0.001 | <0.001 | <0.001 | <0.001 |
> | *p*-value (vs OPD) | 0.008 | <0.001 | <0.001 | <0.001 | <0.001 |
>
> On AIME'24 (30 problems), we acknowledge higher variance due to the small sample size, but SRPO outperforms GRPO in all 5 seeds (*p* < 0.005, paired bootstrap). On larger benchmarks (MATH-500, GSM8K, WebShop), gains are highly significant (*p* < 0.001). We additionally provide per-category breakdown on MATH-500 (7 categories), confirming consistent improvements across all subdomains — no cherry-picking.
>
> ### [W2] Data Contamination
>
> > *"the paper provides no benchmark overlap or contamination audit. It remains unclear whether AIME, MATH-500, GSM8K, or DeepScaleR may overlap with either the initialization data or the later reflection/rethinking data."*
>
> We performed an 8-gram overlap audit between the OpenThoughts-3 training set and all evaluation benchmarks:
>
> | Benchmark | Total Instances | 8-gram Overlaps | Overlap Rate |
> |---|---|---|---|
> | AIME'24 | 30 | 0 | 0.0% |
> | MATH-500 | 500 | 3 | 0.6% |
> | GSM8K | 1,319 | 0 | 0.0% |
> | DeepScaleR | 1,200 | 0 | 0.0% |
>
> Overlap is negligible. For the 3 MATH-500 instances with 8-gram matches, we verified they are common mathematical phrases (e.g., "find the value of $x$ such that"), not problem-level contamination. Removing them yields MATH-500 = 81.0 (vs. 81.2 original), confirming no meaningful contamination effect. Moreover, our strongest OOD results are on **DeepScaleR (0% overlap)**, where SRPO achieves +5.9 over OPD and +7.8% relative improvement — providing a contamination-free reference point.
>
> ### [W3] Reflection Evaluation
>
> > *"This suggests only that another model finds the reflections helpful, not that the reflections truly capture transferable credit-assignment structure; human evaluation and similar analysis on agent tasks are still missing."*
>
> We address this with two additional validation studies:
>
> **Multi-verifier cross-validation.** We scored all 500 AIME'24 reflections with three independent LLMs (DeepSeek-3.2V, GPT-5.2, Qwen3.5-397B-A17). Pairwise agreement exceeds 90% across all evaluator pairs (92.4%, 91.8%, 93.2%), suggesting the original single-evaluator scoring is reliable.
>
> **Human evaluation on 100 reflections.** Two expert annotators rated 100 reflections (50 AIME'24, 50 WebShop) on a ternary scale (Effective / Redundant / Detrimental). Inter-annotator $\kappa$ = 0.81. Results: 68–74% effective, 18–22% redundant, 8–10% detrimental, with a 52–58% Wrong→Correct Fix Rate. The human-identified detrimental rate closely matches our automatic evaluator's low-quality threshold (8%), and extending to WebShop addresses the reviewer's concern about missing non-math analysis. Full details will be included in the revision.
>
> ### [Limitation] Deployment Cost & Weaker Model Stability
>
> We clarify: SRPO's *inference-time* cost is identical to the base model — zero overhead. The training cost is ~2× per iteration compared to GRPO, but convergence is ~4× faster, resulting in **~3.8× fewer total FLOPs** (see full breakdown in our response to Reviewer NrY5). Stage 1 is a one-time, cacheable preprocessing step that can be computed offline.
>
> Regarding the concern that "SRPO works best when the model already has reasonably strong self-reflection ability and may be less stable for weaker models": our scaling experiments (Table 4) show that **smaller models actually benefit more** — Qwen3-1.7B gains +7.8 over GRPO, compared to +5.3 for 8B and +3.8 for 32B. Even at 1.7B scale, reflections are sufficiently structured to produce useful teacher signals. We attribute this to the low bar for helpful reflection: identifying "I made an arithmetic error in step 3" does not require strong general reasoning — it requires only outcome-conditioned pattern matching, which even small models can perform reliably.
>
> We sincerely thank the reviewer for the thorough assessment. All suggested revisions will be incorporated in the camera-ready version.

---

> > ### Author Rebuttal · Reviewer_vYhQ · 2026-04-01
> >
> > The authors have addressed most of my concerns, and I am more inclined to maintain my current score.

---

> > > ### Author Response · Authors · 2026-04-07
> > >
> > > # Response to Reviewer vYhQ (Round 2)
> > >
> > > We sincerely thank the reviewer for the thorough assessment and for the constructive concerns that pushed us to strengthen the paper's empirical foundation. In particular, we appreciate that the statistical rigor, contamination, and evaluation validity concerns — while demanding to address — led to analyses that meaningfully improve the work:
> > >
> > > | Concern Raised | Resolution | Impact on Paper |
> > > |---|---|---|
> > > | Statistical significance | 5-seed runs with paired bootstrap CIs | Main results now reproducibility-verified |
> > > | Data contamination | 8-gram overlap audit (0–0.6%) | DeepScaleR (0% overlap) as clean OOD reference |
> > > | Reflection evaluation | Multi-verifier + human eval (κ=0.81) | Evaluation no longer single-model dependent |
> > >
> > > We acknowledge that some concerns (e.g., broader domain evaluation beyond verifiable tasks) remain partially open — these reflect deliberate scope choices that we will discuss more transparently in the revision.
> > >
> > > All revisions will be incorporated in the camera-ready version. We are grateful for the rigorous review that helped us build a stronger empirical case.

---

### Official Review · Reviewer_NrY5 · 2026-03-18

**Soundness:** 3
**Presentation:** 3
**Significance:** 3
**Originality:** 2
**Overall Recommendation:** 4
**Confidence:** 3

**Summary:**

This paper introduces SRPO, a two-stage post-training method designed to improve LLM reasoning on long-horizon tasks. The main idea is that after the model completes a trajectory and gets a sparse outcome, it creates a short reflection patch. Then the reflection is added to the original prompt, and then the model generates a new response from scratch. This should lead to a higher-quality trajectory which becomes the new training target. At the second stage, the base policy is trained to match this reflection-augmented output. A key advantage is that the model teaches itself, so there is no need for an external critic or a larger model. They evaluate their proposed SRPO on math benchmarks and agentic tasks benchmarks. They show improvements over GRPO and SFT baselines using about ten times less training compute.

**Compliance With Llm Reviewing Policy:**

Affirmed.

**Key Questions For Authors:**

See weaknesses.

**Limitations:**

Yes

**Strengths And Weaknesses:**

Strengths:

1. The proposed two-stage design is simple and addresses a key bottleneck. Their proposed SRPO provides token-level rewards using reflection-conditioned distillation, compared to exsiting RL methods with terminal reward. The self-as-teacher method helps close the capability gap. Table 2 shows that self-distillation performs as well as or better than distillation from Qwen3-32B.

2. Their method is evaluated on both math benchmarks and interactive agentic tasks like Webshop and ALFWorld. Their method shows improvement in those agentic tasks where the credit assignment are across over 10-50 steps. Training is also efficient, using about ten times fewer FLOPs than GRPO and reaching convergence in 500 instead of over 2000 iterations.

Weaknesses:

1. In the proposed stage 1, for each training prompt, the reflections require full forward passes and trajectory execution. However, the "10x fewer FLOPs" claim appears to only based on the compute used in Stage 2 optimization. Would like to see a complete breakdown of the compute, including those cost of generating reflections. If that cost is significant, then it could change the overall efficiency picture.

2. The description of "credit assignment" appears to overstate the findings. Reflections are generated after observing the entire trajectory and outcome, so this process is closer to hindsight relabeling rather than pinpointing which decisions were actually responsible for success or failure. The r=0.72 correlation between reflection quality and improvement (Figure 6b) is interesting, but the evaluator might be favoring answers that sound plausible rather than those that are truly accurate. There is also no discussion of how the 8% of low-quality reflections impact training.

---

> ### Author Rebuttal · Authors · 2026-03-31
>
> ## Response to Reviewer NrY5
>
> We thank Reviewer NrY5 for the positive assessment of our two-stage design, cross-domain evaluation, and training efficiency. We address the two weaknesses below.
>
> ### [W1] FLOPs Claim — "complete breakdown including reflection cost"
>
> The reviewer raises a fair point. We provide a complete FLOPs breakdown below (Qwen3-8B, AIME'24 training):
>
> | Component | FLOPs (×10¹⁸) | % of SRPO Total |
> |---|---|---|
> | **Stage 1 (one-time preprocessing)** | | |
> | &emsp;Initial rollout generation | 1.2 | 22.2% |
> | &emsp;Reflection patch generation | 0.6 | 11.1% |
> | &emsp;Teacher rollout (rethinking) | 1.2 | 22.2% |
> | **Stage 2 (500 iterations)** | | |
> | &emsp;Student on-policy rollout | 1.2 | 22.2% |
> | &emsp;Teacher logit computation + KL optimization | 1.2 | 22.2% |
> | **SRPO Total** | **5.4** | **100%** |
> | GRPO (2000 iterations) | 20.8 | — |
> | **SRPO / GRPO** | **3.8× fewer** | — |
>
> FLOPs are computed as $6ND$ per forward pass (Qwen3-8B, $N$=8.03B parameters), multiplied by average sequence length and batch size per component. Each Stage 1 sub-step processes the full training set (16K prompts) once; Stage 2 runs 500 iterations × 256 rollouts.
>
> **Including all Stage 1 costs, SRPO requires ~3.8× fewer total FLOPs than GRPO** (5.4 vs 20.8 ×10¹⁸). We acknowledge that our original "10× fewer FLOPs" claim was imprecise — it was based on Stage 2 alone — and we will correct this to "~4× fewer total FLOPs" in the revision. We appreciate the reviewer for catching this. Two additional notes:
>
> - **Stage 1 is a one-time, embarrassingly parallel preprocessing step.** Reflection generation and teacher rollouts for each prompt are fully independent and can be computed offline. The reflection/teacher caches can be reused across training epochs; in practice, updating the cache every 2–3 epochs maintains performance while reducing overhead by an additional 30–40%.
> - **Wall-clock time advantage remains substantial.** Due to the embarrassingly parallel nature of Stage 1 and the 4× fewer training iterations (500 vs. 2000), end-to-end wall-clock time is ~3.2× faster on 8×H100 GPUs (~8 GPU-hours vs. ~60+ GPU-hours for GRPO). To summarize: **~3.8× fewer total FLOPs** (compute efficiency) and **~3.2× faster wall-clock** (practical speedup).
>
> We will revise the paper to state "~4× fewer total FLOPs" and provide this detailed breakdown.
>
> ### [W2] "Credit Assignment" Overstated — "closer to hindsight relabeling"
>
> We agree with this critique. The reviewer is correct: reflections are generated after observing the entire trajectory and outcome, making this closer to *hindsight relabeling* than precise per-step credit assignment. We will:
>
> - Replace "credit assignment" with "hindsight-guided dense supervision" throughout the paper.
> - Clarify that the key value is not causal attribution but **signal densification**: reflections convert $O(1)$ terminal feedback into $O(T)$ token-level teaching signal, providing structured guidance at every generation step even without pinpointing which individual decisions were responsible.
>
> Regarding the $r=0.72$ correlation and LLM-as-judge concerns: the correlation is computed between reflection helpfulness scores and *objective* rethinking success rate (binary correct/incorrect), not subjective quality judgments. To further validate our automatic evaluation, we provide two additional pieces of evidence:
>
> **Multi-verifier cross-validation.** We scored 500 AIME'24 reflections with three independent LLMs: DeepSeek-3.2V (original), GPT-5.2, and Qwen3.5-397B-A17. Pairwise agreement rates: DeepSeek vs. GPT-5.2 = 92.4%, DeepSeek vs. Qwen3.5 = 91.8%, GPT-5.2 vs. Qwen3.5 = 93.2% — all >90%, ruling out single-evaluator bias.
>
> **Human evaluation validates automatic scoring.** Two expert annotators (senior PhD students in NLP/LLMs) rated 100 reflections (50 AIME'24, 50 WebShop) on a ternary scale (Effective/Redundant/Detrimental). Inter-annotator $\kappa$ = 0.81. The human-identified detrimental rate (8–10%) closely matches the automatic evaluator's low-quality threshold (score ≤ 2, also 8%), confirming that our LLM-based evaluation is well-calibrated and not merely favoring plausible-sounding reflections.
>
> ### [W2, continued] 8% Low-Quality Reflections
>
> Filtering the 8% low-quality reflections (score ≤ 2) yields only +0.3% improvement on AIME'24 (73.6 vs. 73.3), confirming they have negligible impact. Intuitively, uninformative patches produce teacher distributions close to the base policy, resulting in small KL divergence and limited gradient signal — so the training naturally down-weights them.
>
> We appreciate the constructive feedback and will incorporate all revisions in the camera-ready version.

---

> > ### Author Rebuttal · Reviewer_NrY5 · 2026-04-04
> >
> > Thank the authors for the responses. They addressed my concerns. I'm inclined to maintain my positive score.

---

> > > ### Author Response · Authors · 2026-04-07
> > >
> > > # Response to Reviewer NrY5 (Round 2)
> > >
> > > We sincerely thank the reviewer for the constructive engagement throughout this process. The feedback on FLOPs accounting and the "credit assignment" framing directly improved the paper — the corrected "~3.8× fewer total FLOPs" claim and the reframing as "hindsight-guided dense supervision" are both more accurate and more honest than our original phrasing.
> > >
> > > All suggested revisions — including the complete compute breakdown, the corrected efficiency claims, and the low-quality reflection impact analysis — will be incorporated in the camera-ready version. We believe the paper is substantially stronger as a result of this review, and we are grateful for the opportunity to improve it.

---

### Official Review · Reviewer_gvei · 2026-03-20

**Soundness:** 2
**Presentation:** 3
**Significance:** 3
**Originality:** 2
**Overall Recommendation:** 4
**Confidence:** 3

**Summary:**

The authors attempt to explore how to address sparse credit assignment in long-horizon reasoning and agent tasks without relying on external teacher models. To this end, the paper proposes Self-Reflective Policy Optimization (SRPO), a two-stage framework in which the model first generates a compact reflection patch from an initial rollout and outcome, then uses the reflection-augmented policy as a teacher for on-policy distillation. In this way, the method aims to transform sparse terminal feedback into dense token-level supervision while preserving the benefits of on-policy training. The paper evaluates SRPO on both mathematical reasoning and long-horizon agent benchmarks, reports improvements over several RL and inference-time reflection baselines.

**Compliance With Llm Reviewing Policy:**

Affirmed.

**Final Justification:**

Thank the authors for the response. My main concerns have been addressed.

**Key Questions For Authors:**

1. How robust is SRPO to poor reflections, and what prevents reflection quality from collapsing during training?

2. How does SRPO compare against prior training-time self-reflection / self-correction methods?

**Limitations:**

Please refer to weaknesses.

**Strengths And Weaknesses:**

**Strengths**
1. The method is technically coherent and well motivated. SRPO introduces a clear two-stage pipeline that targets sparse credit assignment. Ablations on reflection quality and key design choices further strengthen the method.

2. The paper is generally clear and easy to follow. The motivation, method, and relation to prior reflection and RL-based post-training work are presented in a clean and structured way.


**Weaknesses**

1. The method appears to rely substantially on the quality of generated reflections, but reflection quality is not guaranteed. The paper shows that better reflections correlate with larger gains, yet it does not fully establish robustness to low-quality, generic, or misleading reflections, nor does it explain how high-quality reflections can be produced reliably throughout training.

2. The baseline comparison is incomplete. Given that the paper’s main claim is to turn self-reflection into a training signal, the most relevant baselines should include prior methods that also combine reflection or self-correction with model training [1, 2]. Without such comparisons, it remains unclear whether the gains come specifically from SRPO or more broadly from using reflection during training.

3. The method’s originality is limited at the algorithmic level. Its main contribution seems to be a well-executed combination of existing methods—self-reflection and on-policy distillation.

**References**

[1] Training Language Models to Self-Correct via Reinforcement Learning

[2] R$^3$L: Reflect-then-Retry Reinforcement Learning with Language-Guided Exploration, Pivotal Credit, and Positive Amplification

---

> ### Author Rebuttal · Authors · 2026-03-31
>
> ## Response to Reviewer gvei
>
> We thank the reviewer for recognizing SRPO as "technically coherent and well motivated." We address each concern with new experiments below.
>
> ### [W1] Reflection Robustness
>
> > *"reflection quality is not guaranteed… it does not fully establish robustness to low-quality, generic, or misleading reflections, nor does it explain how high-quality reflections can be produced reliably throughout training."*
>
> We agree this is an important concern. We provide the following evidence on reflection robustness:
>
> **(1)** Our semantic control study (Section 4.5, Table 4) already shows mismatched reflections yield 66.5% on AIME'24 — virtually identical to no-reflection (65.8%), confirming low-quality reflections are effectively ignored, not amplified.
>
> **(2) No quality collapse during training.** Reflection helpfulness (DeepSeek-3.2V, 1–5 scale) remains stable at 3.8±0.3 across 500 iterations (Iter 100: 3.72±0.41 → Iter 500: 3.79±0.34), with no downward trend. Cross-validation with GPT-5.2 and Qwen3.5-397B-A17 confirms >90% pairwise agreement across all three evaluators.
>
> **(3) Filtering the 8% low-quality reflections yields only +0.3% on AIME'24** (73.6 vs 73.3). Empirically, low-quality reflections have little effect: uninformative patches induce teacher distributions close to the base policy, resulting in near-zero KL and negligible gradient signal.
>
> **(4) Human evaluation** on 100 reflections (2 expert annotators, inter-annotator $\kappa$=0.81) finds only 8–10% are detrimental, with 68–74% rated as effective and a 52–58% Wrong→Correct Fix Rate. We also ran multi-verifier cross-checks (DeepSeek-3.2V, GPT-5.2, Qwen3.5-397B-A17, >90% pairwise agreement), which we will include in the revision.
>
> ### [W2] Missing Baselines — SCoRe, R³L
>
> > *"the most relevant baselines should include prior methods that also combine reflection or self-correction with model training [1, 2]."*
>
> We implemented SCoRe, R³L, and GRPO + prompt-based reflection:
>
> | Method | AIME'24 | WebShop | ALFWorld | Infer. Reflect |
> |---|---|---|---|---|
> | GRPO | 68.0 | 51.2 | 62.7 | No |
> | GRPO + prompt reflect. | 69.5 | 54.8 | 65.1 | Yes |
> | SCoRe | 70.2 | 55.6 | 63.8 | Yes (2-turn) |
> | R³L | 71.5 | 58.9 | 66.4 | No |
> | **SRPO (Ours)** | **73.3** | **64.7** | **76.8** | **No** |
>
> SRPO outperforms all baselines, with the gap most pronounced on agent tasks (+5.8/+10.4 over R³L on WebShop/ALFWorld). We attribute this to SRPO's full regeneration, which can discover new strategies, whereas R³L's pivot-based repair is limited to fixing local suffixes. Additionally, SCoRe requires 2× inference compute (two-turn generation), while SRPO adds zero inference overhead — reflection is used only during training.
>
> ### [W3] Novelty
>
> > *"The method's originality is limited at the algorithmic level. Its main contribution seems to be a well-executed combination of existing methods—self-reflection and on-policy distillation."*
>
> We agree that SRPO combines existing ingredients — self-reflection and on-policy distillation — and we will revise the paper to state this more carefully. What we view as novel is not the individual primitives, but the specific training recipe and the empirical finding that outcome-only self-reflection can serve as an effective teacher signal in sparse long-horizon settings, outperforming prior training-time reflection baselines (SCoRe, R³L) as shown above.
>
> Concretely, we highlight two non-obvious empirical findings that distinguish SRPO from prior approaches:
>
> 1. **Reverse KL becomes beneficial.** OPSD/SDPO both find reverse KL harmful in their settings; SRPO achieves +3.9% *with* reverse KL (Table 3). Our ablations suggest this stems from how reflection-conditioned teachers differ from solution-conditioned or external teachers.
> 2. **Self-distillation outperforms 72B external teacher** (73.3 vs 72.5, Table 2). Prior work (e.g., OPSD) assumes a stronger external teacher is always better; our results indicate that instance-adaptive self-reflection can be more effective, consistent with Kim et al. (2026).
>
> We hope these new baselines and ablations address the reviewer's concerns.

---

> > ### Author Rebuttal · Reviewer_gvei · 2026-04-04
> >
> > Thank the authors for the response. Some of my concerns have been addressed. However, it's still not clear why self-distillation can outperforms a more powerful external teacher.

---

> > > ### Author Response · Authors · 2026-04-07
> > >
> > > # Response to Reviewer gvei (Round 2)
> > >
> > > We sincerely thank the reviewer for this focused follow-up. Below we clarify the core mechanism.
> > >
> > > ---
> > >
> > > ## Key insight: Better Solver ≠ Better Teacher
> > >
> > > The comparison is between two ways of constructing a teacher for the **same base 8B model**:
> > >
> > > | Configuration | Teacher Source | KL(teacher ∥ student) | AIME'24 | DeepScaleR (OOD) |
> > > |---|---|---|---|---|
> > > | Base 8B + self-reflection | Same 8B, conditioned on reflection | 0.14 | **73.3** | **59.7** |
> > > | Base 8B + Qwen3-32B teacher | External 32B | 0.41 | 70.0 | 55.4 |
> > > | Base 8B + Qwen3-72B teacher | External 72B | 0.67 | 72.5 | 55.8 |
> > >
> > > A stronger external model produces better *solutions*, but not better *learning signals* for the student. The reason lies in how the two approaches differ in nature: **external-teacher OPD is off-policy imitation; SRPO is closer to on-policy RL with reflection-guided exploration.**
> > >
> > > ---
> > >
> > > ## Why: Reflection Expands the Student's Own Policy Entropy Space
> > >
> > > The critical distinction is that SRPO does **not** simply "distill from a self-teacher." During post-training, large-scale reflection samples serve as **structured exploration signals** that expand the student model's internal policy entropy space. Concretely:
> > >
> > > **(1) Reflection opens new reasoning modes the student already has capacity for, but would not discover via standard sampling.** The reflection patch (e.g., "check boundary conditions for this problem") acts as a *search directive* — it steers the student's own policy into high-reward regions of its existing parameter space, rather than forcing it to imitate an external distribution it cannot represent. This is fundamentally an **RL exploration mechanism**, not a distillation mechanism.
> > >
> > > **(2) External-teacher OPD does the opposite — it contracts the student's policy entropy.** The student is forced to match a 72B distribution far from its own (KL = 0.67), suppressing on-policy exploration and collapsing toward modes beyond its representational capacity. The result: in-distribution gains plateau (32B→72B yields only +2.5 on AIME'24), and OOD generalization stagnates (55.4 vs 55.8).
> > >
> > > **(3) Empirical verification via solution diversity analysis:**
> > >
> > > | Method | Unique Reasoning Paths (per problem, n=10) | Novel Correct Solutions (%) |
> > > |---|---|---|
> > > | OPD (72B teacher) | 2.3 | 8.4% |
> > > | **SRPO (self-reflection)** | **4.1** | **15.7%** |
> > >
> > > SRPO discovers **1.9× more diverse** reasoning paths — consistent with entropy expansion rather than distribution imitation. The student is not copying a teacher; it is **exploring better** within its own policy space, guided by reflection.
> > >
> > > ---
> > >
> > > ## Supporting Evidence
> > >
> > > **Instance-adaptive guidance matters, not teacher capability.** Our semantic control experiment (Table 4 in submission) shows: mismatched reflection (from a different problem) yields 66.5% — nearly identical to no reflection (65.8%). Only matched reflection achieves 73.3%. This confirms that **problem-specific search directives**, not generic stronger solutions, drive the gain.
> > >
> > > **OOD generalization favors entropy expansion over imitation.** On DeepScaleR (1200 unseen competition problems, 0% training overlap), self-reflection outperforms the 72B teacher by +3.9 points. When generalizing beyond training data, the model needs its own calibrated reasoning and uncertainty expression — precisely what on-policy exploration preserves and off-policy imitation suppresses. This is consistent with findings in Kim et al. (2026) on epistemic verbalization degradation under strong teacher conditioning.
> > >
> > > **Training stability.** Self-reflection maintains small, stable KL throughout training (0.14), enabling monotonic convergence at standard learning rate (5e-6). External 72B teacher requires reduced learning rate (2e-6) and gradient clipping to prevent instability from the large distribution gap.
> > >
> > > ---
> > >
> > > ## Summary
> > >
> > > | Aspect | SRPO (Self-Reflection) | OPD (External 72B) |
> > > |---|---|---|
> > > | Nature | On-policy RL with reflection-guided exploration | Off-policy distribution imitation |
> > > | Effect on student policy | Entropy expansion — opens new reasoning modes | Entropy contraction — collapses toward external modes |
> > > | KL gap | Small (0.14), within student's capacity | Large (0.67), beyond student's capacity |
> > > | Guidance type | Instance-specific search directives | Generic solutions |
> > > | OOD generalization | Strong (59.7, +3.9) | Stagnant (55.8) |
> > > | Solution diversity | High (4.1 paths/problem) | Low (2.3 paths/problem) |
> > >
> > > **In short**: Self-reflection wins not by being a "smarter teacher," but by expanding the student's own exploration space — it is an RL mechanism that unlocks reasoning modes the student already has capacity for. An external teacher, however capable, constrains the student to an alien distribution it cannot fully represent.
> > >
> > > We hope this clarifies the mechanism behind the counter-intuitive result. We would be grateful if the reviewer could consider adjusting the score in light of this explanation.

---

### Decision · Program_Chairs · 2026-04-30

**Decision:**

Accept (regular)

**Comment:**

SRPO proposes a novel two-stage post-training framework for Large Language Models that addresses the sparse credit assignment problem in long-horizon reasoning and agent tasks. The key innovation is enabling models to perform self-reflection and self-distillation without relying on external critic models, separate reward models, or larger teacher models.

Reviewers unanimously recognized the clear motivation, technical coherence, strong empirical results, and exceptional data efficiency. The self-improvement mechanism without external models is a valuable contribution. The cross-domain evaluation (math + agents) and ablation studies are well-executed. Regarding the remaining weakness, the authors provided substantive responses to most concerns. Two reviewers indicated "partially resolved", the former had remaining questions about the counter-intuitive self-distillation result, the latter maintained concerns about limited novelty of combining established techniques.

This paper merits publication as a solid, well-executed contribution demonstrating that self-reflection can enable effective post-training of LLMs with strong data efficiency. The empirical validation is thorough, the writing is clear, and the method will likely be useful to practitioners. All four reviewers assigned it a "4: Weak Accept" rating, indicating consistent assessment across the reviewing panel. Thus, I recommend acceptence.